# Mamba-IVP: A Denoising State–Space Initial Value Problem Framework for SOTA Clinical Time Series, Healthcare Alternative

## Abstract

Missing clinical time series is a critical bottleneck in intensive care units (ICUs). In large-scale ICU electronic health record datasets such as MIMIC-IV, missing rates exceed 90% due to sensor failures, monitor degradation, and systemic outages, while aging devices inject unstable noise that makes reliable modeling nearly impossible. Existing methods remain unsafe for deployment: statistical heuristics distort missingness, deep models collapse under block-wise gaps and noise, and ODE- or diffusion-based approaches demand prohibitive computation. To overcome these limitations, we propose Mamba-IVP, a state–space generative model with a Mask-Aware Dual-Mamba Encoder (MADME) to handle block-wise missingness and a Mamba-Hybrid Decoder (MHD) to denoise continuous-time reconstructions. We validate our method through 61 experiments across two tasks: time series forecasting and node classification. Our experiments involve 7 classic and state-of-the-art target models and 3 publicly available datasets: (1) it achieves state-of-the-art accuracy, reducing MSE by 3.0%, improving AUROC by 3.0%, and enhancing AUPRC by 3.9%; and (2) it remains robust under noise and block-wise missingness up to 12h, where other models degrade sharply.

## 1 Introduction

Reliable prediction in the ICU is not just an algorithmic challenge but a matter of life and death. Electronic health records (EHRs) and continuous bedside monitoring hold the promise of enabling early detection of disease trajectories, timely intervention, and improved survival. Yet this promise is routinely shattered by the brutal reality of clinical data: incompleteness, irregularity, and noise. In large-scale datasets such as MIMIC-IV, missing rates exceed 90%, with block-wise missingness of 2–6 hours, and in extreme cases, up to 12 hours Johnson et al. (2016). These gaps are not harmless, and the consequence is catastrophic. Every gap in the data translates into lost lives, turning missingness into serious consequences in ICU operations. Addressing this crisis is therefore not a technical preference but a life-or-death imperative for building trustworthy, deployable healthcare AI.

Existing approaches to modeling incomplete clinical time series can be broadly categorized into three methodological paradigms, each facing critical limitations for real-world deployment. The first paradigm, the *imputation–then–prediction* pipeline (e.g., MissForest + GRU Stekhoven & Bühlmann (2012), SAITS + classifier Oh et al. (2021)), reconstructs missing values prior to downstream prediction. However, imputation errors often compound through the pipeline, amplifying uncertainty, while the two-stage design doubles computational overhead and latency. The second paradigm comprises *end-to-end models with built-in missingness handling* (e.g., GRU-D Che et al. (2018c), BRITS Cao et al. (2018b), mTAN Shukla & Marlin (2021)), which directly integrate masks or decay mechanisms into recurrent or attention-based architectures. Although these models circumvent explicit imputation, they typically assume independent or random missingness patterns and degrade sharply under the structured, block-wise gaps commonly observed in ICU monitoring.

A third line of research introduces *continuous-time generative models* (e.g., Latent-ODE Rubanova et al. (2019c), IVP-VAE Xiao et al. (2024b)) that model irregular sampling through latent dynamical systems. While elegant in theory, these approaches face several practical obstacles: computational

bottlenecks due to adaptive ODE solvers Chen et al. (2018b) (often 40× slower than our proposed method), sensitivity to measurement noise and sensor drift, and the entanglement of observation patterns with data content, which undermines robustness and interpretability. These limitations collectively reveal the urgent need for a unified, computationally efficient, and noise-resilient framework capable of learning stable temporal dynamics from irregular, incomplete clinical sequences.

To directly address these challenges, we propose Mamba-IVP, a generative framework purpose-built for irregular, noisy, and long-range missing clinical series. First, to confront the challenge of block-wise missingness, we design a Mask-Aware Dual-Mamba Encoder (MADME) that jointly encodes observed values and missingness indicators, ensuring the model learns temporal dynamics robustly even when entire time blocks vanish. Second, to mitigate measurement noise from aging devices, we introduce a Mamba-Hybrid Decoder (MHD) that reconstructs continuous-time trajectories while inherently denoising through parallelizable latent evolution. Third, to overcome the prohibitive computational cost of ODE-, diffusion-, and flow-based models, Mamba-IVP leverages Mamba's parallelizable state–space dynamics, eliminating recursive bottlenecks and enabling efficient sequence modeling. Furthermore, by embedding the encoder–decoder pair within an invertible solver, our framework preserves temporal consistency while supporting scalable, real-time inference.

Our main contributions are summarized as follows:

1. We propose a Mask-Aware Dual-Mamba Encoder (MADME) that jointly processes values and missingness indicators to learn robust temporal representations under block-wise missingness. Leveraging Mamba's efficient state–space sequence modeling, MADME substantially improves stability under long gaps and achieves strong forecasting performance (MSE = 0.697 on MIMIC-IV, 0.544 on PhysioNet 2012, and 0.564 on eICU). These correspond to **3–4% lower MSE than IVP–VAE** in the main forecasting tasks, and **up to 7.3%** improvement compared to GRU-$\Delta_t$ and IVP–VAE under 10h block-wise missingness.

2. We develop a Mamba-Hybrid Decoder (MHD) that reconstructs continuous-time trajectories while inherently denoising through parallelizable evolution. By combining state–space refinement with lightweight feedforward decoding, MHD enhances robustness to noisy and irregular measurements. On PhysioNet 2012, it yields **4% lower MSE than IVP–VAE** in the main task and **up to 7.3% improvement** under block-wise missingness. In addition, MHD contributes to the substantial speed gains reported in Section 5.6.

3. We provide the first rigorous theoretical analysis of Mamba's denoising power. Our variance-contraction results show that clean tokens induce *exponential* error contraction, while noisy or masked tokens exhibit only *linear* error growth. This analysis explains why the shared IVP solver remains stable and why Mamba-IVP maintains temporal consistency when evolving both backward and forward in time, even under long missing blocks.

4. Extensive experiments demonstrate that Mamba-IVP achieves the best accuracy–efficiency trade-off among all baselines. It attains the lowest forecasting MSE across datasets (e.g., 0.544 on PhysioNet 2012, **3–4% better than IVP–VAE**), while providing the **fastest forward time** (0.007s) and **shortest epoch time** (5.2s). Compared to the computationally heavy Latent-Flow baseline, Mamba-IVP is **up to 40× faster** in inference. Even under 50% masking noise, it maintains strong robustness (MSE = 0.709), achieving **over 40% lower MSE than IVP–VAE** in our robustness experiments.

## 2 RELATED WORK

Due to space limits, we briefly review representative work and include an extended survey in Appendix A.14. Early imputation relied on statistical methods such as MICE Van Buuren & Groothuis-Oudshoorn (2011), 3D-MICE Xu et al. (2023), and TA-DualCV Zhang & Thorburn (2021) that capture conditional or spatiotemporal dependencies but fail under ICU-level sparsity and long block-wise gaps. Deep learning methods including GRU-D Che et al. (2018c), BRITS!Cao et al. (2018a), SAITS Oh et al. (2021), and diffusion- or state-space-based models (CSDI Tashiro et al. (2021c), GRIN!García-Recio et al. (2021), diffusion-SSM Oh et al. (2021)) improve temporal modeling yet remain computationally intensive and focus mainly on reconstruction accuracy. Our **Mamba-IVP** differs by integrating a mask-aware Mamba encoder and IVP-based decoder to jointly address block-wise missingness and noisy observations for both imputation and downstream prediction.

## 3 PRELIMINARY

### 3.1 PROBLEM FORMULATION

In our framework, a multivariate time series $X^{(n)}$ is defined as a sequence of $L_n$ temporally ordered observations:

$$X^{(n)} = \{(\mathbf{x}_i^{(n)}, t_i^{(n)})\}_{i=1}^{L_n},$$

where each $\mathbf{x}_i^{(n)} \in \mathbb{R}^D$ is a $D$-dimensional feature vector (e.g., vital signs in EHRs), and $t_i^{(n)} \in \mathbb{R}^+$ is the timestamp. The sequence length $L_n$ varies across patients due to irregular sampling.

The dataset $\mathbf{X} = \{(X^{(1)}, y^{(1)}), \ldots, (X^{(N)}, y^{(N)})\}$ contains $N$ labeled samples, where $X^{(n)}$ can be represented by a feature matrix $\mathbf{X}^{(n)} \in \mathbb{R}^{L_n \times D}$ with timestamps $\mathbf{t}^{(n)} \in \mathbb{R}^{L_n}$ and label $y^{(n)} \in \mathcal{Y} = \{1, \ldots, C\}$. All sequences lie within a study window $[T_{\text{start}}, T_{\text{end}}]$, but each patient terminates at its own endpoint $T_{\text{obs}}^{(n)} := t_{L_n}^{(n)}$.

**Definition 1** (Problem Statement). We aim to design a unified generative model $g_\theta$, parameterized by $\theta$, that learns a common latent representation from irregular time series. Once trained, the model supports multiple downstream tasks via distinct inference pathways.

For clarity of exposition, we focus on a single input sample in the following discussion and omit the superscript $(n)$ unless ambiguity arises.

- **Time Series Forecasting:** Given a historical time series $X = \{(\mathbf{x}_i, t_i)\}_{i=1}^{L}$ observed over the interval $[T_{\text{start}}, T_{\text{obs}}]$, observed up to its own endpoint $T_{\text{obs}} := t_L$, predict the future sequence

$$\hat{X}^\tau = \{(\hat{\mathbf{x}}_{L+k}, t_{L+k})\}_{k=1}^{L_\tau},$$

where each forecasted timestamp satisfies

$$t_{L+k} \in (T_{\text{obs}}, T_{\text{obs}} + \tau], \qquad t_{L+L_\tau} \leq T_{\text{end}} + \tau,$$

and $\tau \in \mathbb{R}^+$ is the prediction horizon.
- **Time Series Classification:** The task is to infer the categorical label $\hat{y} \in \mathcal{Y}$ corresponding to the entire time series $X$.

*Comment 1:* The generative model $g_\theta$ serves as a representation learner, extracting temporally coherent features that simultaneously support forecasting and classification.

*Comment 2:* By reusing temporal dependencies captured during forecasting, the framework enhances discriminative accuracy, creating a synergistic link between generative modeling and classification.

*Comment 3:* Unlike traditional pipelines that separate imputation, forecasting, and classification, our latent trajectory framework unifies them: bidirectional evolution reconstructs missing values for complete representations, while forward extrapolation enables forecasting under high missingness.

## 4 METHOD

Our framework consists of three tightly connected components: (1) a **Mask-Aware Dual-Mamba Encoder (MADME)** that processes irregular, partially observed inputs; (2) a **bidirectional latent evolution module** based on an initial value problem (IVP) solver, which evolves the encoded representations backward and forward in continuous time; and (3) a **Mamba-Hybrid Decoder (MHD)** that reconstructs or forecasts observations from the forward latent trajectory.

The overall workflow is as follows: the input values $\mathbf{X}$ and observation mask $\mathbf{M}$ are concatenated and encoded by MADME into a latent sequence $\mathbf{Z}$. We then evolve this sequence backward in time (EBT) to obtain a temporally coherent summary, which is aggregated into a compact latent representation $\hat{\mathbf{z}}_{\text{init}}$. This aggregated state is used for classification and simultaneously serves as the initial condition for a forward-time latent evolution (EFT) that generates $\mathbf{Z}^\rightarrow$ at future timestamps. Finally, MHD refines this trajectory using Mamba-style state–space dynamics and projects it to predicted observations $\hat{\mathbf{X}}$.

We next describe each component in the order in which data flows through the system.

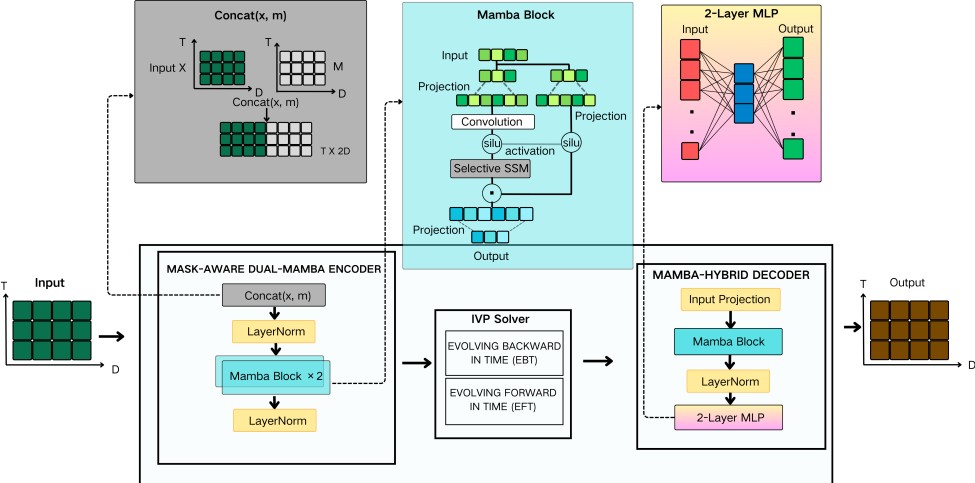

Figure 1: Overview of the Mamba-IVP framework. Multivariate observations $\mathbf{X}$ and their binary mask $\mathbf{M}$ are concatenated and encoded by the **Mask-Aware Dual-Mamba Encoder (MADME)** into latent embeddings $\mathbf{Z}$. These are evolved *backward* via a shared IVP solver to obtain a trajectory $\mathbf{Z}^{\leftarrow}$, which is aggregated into a compact latent state $\hat{\mathbf{z}}_{\text{init}}$ for classification and as the initial condition for *forward* evolution, producing a future latent trajectory $\mathbf{Z}^{\rightarrow}$. The **Mamba-Hybrid Decoder (MHD)** then maps $\mathbf{Z}^{\rightarrow}$ to predicted observations $\hat{\mathbf{X}}$. This encoder–IVP–decoder pipeline is designed to handle irregular sampling, block-wise missingness, and sensor noise in clinical time series.

## 4.1 MASK-AWARE DUAL-MAMBA ENCODER (MADAE)

The overall architecture of our model is illustrated in Figure 1, and the algorithm is presented in Appendix A.2.

Clinical time series exhibit both irregular sampling and block-wise missingness, making it crucial for the encoder to distinguish between real measurements and unobserved entries. MADME addresses this by jointly processing raw values and their binary observation mask. Specifically, the input is first formed as

$$\tilde{\mathbf{X}} = \text{Concat}(\mathbf{X}, \mathbf{M}) \in \mathbb{R}^{L \times 2D}. \tag{1}$$

ensuring that the encoder is explicitly aware of missing regions at every time step.

MADME then applies two stacked Mamba blocks (see Appendix A.1) with residual connections. These blocks model long-range temporal dependencies while selectively filtering noisy or unreliable inputs.

$$\mathbf{H}^0 = \text{LayerNorm}(\tilde{\mathbf{X}}) \cdot \mathbf{W}_{\text{proj}} + \mathbf{b}_{\text{proj}}, \tag{2}$$

$$\mathbf{H}^1 = \mathbf{H}^0 + \text{Mamba}_1\left(\mathbf{H}^0\right), \tag{3}$$

$$\mathbf{H}^2 = \mathbf{H}^1 + \text{Mamba}_2\left(\text{Dropout}\left(\mathbf{H}^1\right)\right). \tag{4}$$

The initial representation is denoted by $\mathbf{H}^0 \in \mathbb{R}^{L \times d_m}$, obtained through layer normalization followed by a learnable linear projection defined by $\mathbf{W}_{\text{proj}}$. The projection weights and bias are $\mathbf{W}_{\text{proj}} \in \mathbb{R}^{2D \times d_m}$ and $\mathbf{b}_{\text{proj}} \in \mathbb{R}^{d_m}$, respectively. The output of the first Mamba block with a residual connection is $\mathbf{H}^1 \in \mathbb{R}^{L \times d_m}$, and the final encoded sequence after the second Mamba block and dropout is denoted as $\mathbf{H}^2 \in \mathbb{R}^{L \times d_m}$. The residual connections help stabilize training and retain input features, while each $\text{Mamba}_i(\cdot)$ module models long-range temporal dependencies through efficient state space representations.

Finally, we normalize and project the encoded sequence into a latent trajectory space:

$$\mathbf{Z} = \text{LayerNorm}(\mathbf{H}^2) \cdot \mathbf{W}_{\text{out}} + \mathbf{b}_{\text{out}}, \quad \mathbf{Z} \in \mathbb{R}^{L \times d_z}, \tag{5}$$

where the final output of the encoder is denoted as $\mathbf{Z} \in \mathbb{R}^{L \times d_z}$, where each row vector $\mathbf{z}_t \in \mathbb{R}^{d_z}$ represents the latent representation at time step $t$. The output projection uses a learnable weight

matrix $\mathbf{W}_{\text{out}} \in \mathbb{R}^{d_m \times d_z}$ and a bias term $\mathbf{b}_{\text{out}} \in \mathbb{R}^{d_z}$. The dimensionality of the final latent space is denoted by $d_z$.

## 4.2  EVOLVING BACKWARD IN TIME (EBT)

To extract a compact latent summary from partially observed time series, we simulate a latent trajectory backward in time, inspired by Xiao et al. (2024a). This backward simulation addresses the problem of missing values by allowing the model to infer a globally coherent latent representation without explicit imputation. Instead of filling in the missing entries, we directly encode the observation mask and learn to model latent dynamics conditioned on partial observations.

Specifically, given the temporally contextualized embeddings $\mathbf{Z} = [\mathbf{z}_1, \ldots, \mathbf{z}_L] \in \mathbb{R}^{L \times d_z}$ from our Mask-Aware Dual-Mamba Encoder and their associated timestamps $\mathbf{t}_{\text{in}} = [t_1, \ldots, t_L] \in \mathbb{R}^L$, we reverse the time axis as $\mathbf{t}_{\text{rev}} = [t_L, \ldots, t_1]$ and use the last embedding $\mathbf{z}_L$ as the initial latent state.

We define a neural initial value problem (IVP) solver to simulate the latent evolution by solving the following ordinary differential equation with initial conditions:

$$\frac{d\mathbf{z}(t)}{dt} = f_\theta(\mathbf{z}(t), t), \quad \mathbf{z}(t = t_L) = \mathbf{z}_L, \tag{6}$$

Here, $t$ denotes a continuous time variable used by the ODE solver, sampled from the reversed time vector $\mathbf{t}_{\text{rev}}$. This formulation allows us to simulate the latent dynamics backward from the final timestamp $t_L$. The function $f_\theta(\cdot, \cdot)$ represents a learnable neural module parameterizing the latent dynamics. In our experiments, we instantiate $f$ as either a multi-layer perceptron (MLP) or a residual neural flow (ResNetFlow), each used in separate runs under a unified solver interface.

The IVP solver then generates a reverse-evolved latent trajectory:

$$\mathbf{Z}^{\leftarrow}(t) = \text{IVPSolver}\left(f, \mathbf{z}_L, \mathbf{t}_{\text{rev}}\right), \quad \mathbf{Z}^{\leftarrow}(t) \in \mathbb{R}^{L \times d_z}, \tag{7}$$

where $\mathbf{Z}^{\leftarrow}(t)$ denotes the latent trajectory inferred along the reversed time axis. Here, $t$ corresponds to the reversed time values in $\mathbf{t}_{\text{rev}}$, and thus the initial time of the integration is $t_L = T_{\text{obs}}$, proceeding backward to $T_{\text{start}}$.

To summarize this trajectory into a compact latent representation, we apply an aggregation function Aggregate$(\cdot)$ over all valid time steps. Depending on the downstream task, this function can be instantiated as a simple weighted average, a learned attention mechanism, or a KL-divergence-based selector. The aggregated result is:

$$\hat{\mathbf{z}}_{\text{init}} = \text{Aggregate}\left(\mathbf{Z}^{\leftarrow}(t)\right), \quad \hat{\mathbf{z}}_{\text{init}} \in \mathbb{R}^{d_z}, \tag{8}$$

where $\hat{\mathbf{z}}_{\text{init}}$ is a compact vector summarizing the entire observed history. Since it is a global latent representation rather than a sequence, we use lowercase $\mathbf{z}$ to emphasize its non-temporal nature. It lies in the $d_z$-dimensional latent space, is a vector in $\mathbb{R}^{d_z}$.

This backward evolution allows the model to integrate temporally local embeddings into a globally coherent latent state, which serves as the initial condition for modeling future dynamics.

### 4.2.1  EBT FOR CLASSIFICATION

Following the backward trajectory evolution described above, we obtain a globally aggregated latent representation $\hat{\mathbf{z}}_{\text{init}}$, which summarizes the observed history of the sample. For binary classification tasks, we directly use this latent vector as input to a simple classifier:

$$\hat{y} = \sigma\left(\text{MLP}_{\text{clf}}\left(\hat{\mathbf{z}}_{\text{init}}\right)\right), \tag{9}$$

where $\text{MLP}_{\text{clf}}$ is a feedforward neural network and $\sigma(\cdot)$ denotes the sigmoid function. The output $\hat{y} \in (0, 1)$ represents the predicted probability of the positive class.

Since $\hat{\mathbf{z}}_{\text{init}}$ originates from backward IVP evolution and aggregates latent signals across the observation window, it encodes both temporal dependencies and missingness patterns in a compact form. This enables the classifier to make label predictions without requiring access to the original time series or explicit imputation.

## 4.3 Evolving Forward in Time (EFT)

Given the aggregated latent initial state $\hat{\mathbf{z}}_{\text{init}} \in \mathbb{R}^{d_z}$ obtained from EBT, we simulate the latent dynamics forward over a target prediction axis $\mathbf{t}_{\text{out}} = [t_{L+1}, \ldots, t_{L+L_\tau}] \in \mathbb{R}^{L_\tau}$, which spans the interval immediately following the last observed timestamp $t_L$ in $\mathbf{t}_{\text{in}} = [t_1, \ldots, t_L]$. This vector is constructed during preprocessing and defines the temporal horizon for prediction.

The latent trajectory is then evolved forward in time using the same neural initial value problem (IVP) solver introduced earlier:

$$\mathbf{Z}^{\rightarrow}(t) = \text{IVPSolver}\left(f_\theta, \hat{\mathbf{z}}_{\text{init}}, \mathbf{t}_{\text{out}}\right), \quad \mathbf{Z}^{\rightarrow}(t) \in \mathbb{R}^{L_\tau \times d_z}, \tag{10}$$

where we instantiate $f_\theta$ as either a multi-layer perceptron (ODE) or a residual neural flow (ResNet-Flow), each used in separate runs under a unified solver interface. The output sequence is represented as $\mathbf{Z}^{\rightarrow}(t) = [\mathbf{z}_{L+1}^{\rightarrow}, \ldots, \mathbf{z}_{L+L_\tau}^{\rightarrow}]$, corresponding to the timestamps in $\mathbf{t}_{\text{out}}$.

The resulting latent sequence serves as input to the Mamba-Hybrid decoder, which maps the latent states to the predicted future observations in the original data space.

## 4.4 Mamba-Hybrid Decoder (MHD)

To address the challenge of measurement noise from aging and unstable ICU devices, we design the Mamba-Hybrid Decoder (MHD). Its goal is to reconstruct continuous-time trajectories while inherently denoising through parallelizable latent evolution. Given the latent trajectory $\mathbf{Z}^{\rightarrow}(t) = [\mathbf{z}_{L+1}^{\rightarrow}, \ldots, \mathbf{z}_{L+L_\tau}^{\rightarrow}] \in \mathbb{R}^{L_\tau \times d_z}$ obtained from the encoder and latent solver, where each $\mathbf{z}_{t_{L+k}}^{\rightarrow} \in \mathbb{R}^{d_z}$ represents the latent state at time $t_{L+k}$, the decoder processes these states with Mamba-based state–space dynamics to generate the predicted observation sequence. This design enables robust denoising, accurate temporal reconstruction, and scalable inference.

First, the latent sequence is projected into a decoder feature space of dimension $d_m$:

$$\mathbf{H} = \mathbf{Z}^{\rightarrow}(t) \cdot \mathbf{W}_{\text{in}}^{\top} + \mathbf{b}_{\text{in}}, \quad \mathbf{H} \in \mathbb{R}^{L_\tau \times d_m}, \tag{11}$$

where $\mathbf{W}_{\text{in}} \in \mathbb{R}^{d_m \times d_z}$ and $\mathbf{b}_{\text{in}} \in \mathbb{R}^{d_m}$ are learnable parameters. Let $\mathbf{H} = [\mathbf{h}_{L+1}, \ldots, \mathbf{h}_{L+L_\tau}]$ denote the sequence of hidden decoder features, where each $\mathbf{h}_{L+k} \in \mathbb{R}^{d_m}$ corresponds to the projected latent embedding at time $t_{L+k}$.

To capture temporal dependencies, we apply a discrete-time Mamba state–space block (as described in Appendix A.1) across the entire sequence:

$$\hat{\mathbf{H}} = \text{LayerNorm}(\text{Mamba}(\mathbf{H}) + \mathbf{H}), \quad \hat{\mathbf{H}} \in \mathbb{R}^{L_\tau \times d_m}, \tag{12}$$

where $\text{Mamba}(\cdot)$ denotes the sequential modeling module and the residual connection ensures stable gradient propagation. The output $\hat{\mathbf{H}} = [\hat{\mathbf{h}}_{L+1}, \ldots, \hat{\mathbf{h}}_{L+L_\tau}]$ contains the temporally-refined hidden representations at each prediction step.

Each Mamba-refined hidden state $\hat{\mathbf{h}}_{t_{L+k}}$ is then passed through a two-layer multilayer perceptron (MLP) to produce the predicted future observation:

$$\hat{\mathbf{X}} = \text{ReLU}(\hat{\mathbf{H}} \cdot \mathbf{W}_1^{\top} + \mathbf{b}_1) \cdot \mathbf{W}_2^{\top} + \mathbf{b}_2, \quad \hat{\mathbf{X}} \in \mathbb{R}^{L_\tau \times D}, \tag{13}$$

where $\mathbf{W}_1 \in \mathbb{R}^{d_h \times d_m}$, $\mathbf{b}_1 \in \mathbb{R}^{d_h}$, $\mathbf{W}_2 \in \mathbb{R}^{D \times d_h}$, and $\mathbf{b}_2 \in \mathbb{R}^D$ are learnable parameters. Let $\hat{\mathbf{X}} = [\hat{\mathbf{x}}_{L+1}, \ldots, \hat{\mathbf{x}}_{L+L_\tau}]$, where each $\hat{\mathbf{x}}_{L+k} \in \mathbb{R}^D$ is the predicted observation at future time $t_{L+k}$. In this way, we have completed the prediction and obtained the final prediction result.

## 4.5 The Denoise Power of Mamba

Consider the selective state—space (Mamba) update Gu & Dao (2024)

$$g_t = \sigma(W x_t), \qquad h_t = (1 - g_t) h_{t-1} + g_t x_t, \tag{14}$$

where $x_t = m_t s_t$ is the observed token, $s_t$ the clean signal, $\{m_t\}_{t \geq 0}$ the mask. Throughout the paper, we assume that $m_t$ is independent of $\{h_\tau, s_\tau, n_\tau\}_{\tau < t}$.

We do not fix a distribution for $m_t$, we attempt to give a general form of the denoise mamba, therefore, we do not fix the noise distribution, we only assume the first two moments exist $\mu := \mathbb{E}[m_t]$ and $\sigma_m^2 := \mathrm{Var}[m_t] < \infty$. The clean signal is bounded, $\sup_{t \geq 0} |s_t| \leq S < \infty$. Then the zero-mean noise and a uniform variance bound can be written as the centred noise

$$n_t := x_t - \mathbb{E}[x_t] = (m_t - \mu)\, s_t,$$

so that $\mathbb{E}[n_t] = 0$ and $\mathrm{Var}[n_t] = \sigma_m^2\, s_t^2 \leq \sigma_m^2\, S^2 =: \sigma_n^2$. Hence $\mathrm{Var}[n_t] \leq \sigma_n^2$ *uniformly in* $t$.

For the Gate constants $0 < \eta_{noise} < \eta_{clean} < 1$ such that the data–driven gate $g_t$ satisfies

$$\begin{cases} g_t \geq \eta_{clean} & \text{(reliable / clean token)}, \\ g_t \leq \eta_{noise} & \text{(noised / corrupted token)}. \end{cases}$$

Because $g_t = \sigma(Wx_t)$ depends on the random mask $m_t$, $g_t$ is itself a random variable. The inequalities $g_t \geq \eta_{clean}$ (clean) and $g_t \leq \eta_{noise}$ (masked) are assumed to hold almost surely. Consequently, for every realisation we have $(1 - g_t)^2 \leq (1 - \eta_{clean})^2$ or $g_t^2 \leq \eta_{noise}^2$, so the variance bounds that follow are path-wise valid. The two regimes are analysed separately below.

**Lemma 1** (Variance contraction on clean tokens)**.** *If $g_t \geq \eta_{clean}$, then*

$$\mathrm{Var}[h_t] \leq (1 - \eta_{clean})^2\, \mathrm{Var}[h_{t-1}] + \sigma_n^2.$$

*Proof.* (Detailed proof is in Appendix A.3) since the bound is $g_t^2 \sigma_n^2 \leq 1 \cdot \sigma_n^2$, so we drop 1. □

To extend the single time step with further $L$ steps, the variance contraction can be defined as an exponential stabilisation.

**Corollary 1** (Exponential stabilisation)**.** *If a run of $L$ consecutive tokens satisfies $g_{t+\ell} \geq \eta_{clean}$ ($\ell = 0, \ldots, L-1$), then*

$$\mathrm{Var}[h_{t+L}] \leq (1 - \eta_{clean})^{2L}\, \mathrm{Var}[h_t] + \frac{1 - (1 - \eta_{clean})^{2L}}{1 - (1 - \eta_{clean})^2}\, \sigma_n^2.$$

For the masked/noised tokens (the noise case), when the Mamba gate is properly bounded (which is naturally achieved through sigmoid activations or learned constraints), the variance of hidden states cannot explode even when processing masked/noised tokens:

**Lemma 2** (No blow-up on masked/noised tokens)**.** *If $g_t \leq \eta_{noise}$, then*

$$\mathrm{Var}[h_t] \leq \mathrm{Var}[h_{t-1}] + \eta_{noise}^2\, \sigma_n^2.$$

*Proof.* Using the same variance expression, $(1 - g_t)^2 \leq 1$ and $g_t^2 \leq \eta_{noise}^2$ when $g_t \leq \eta_{noise}$. (Detailed proof is in the Appendix A.4) □

We now extend our analysis to characterize the cumulative effect of processing multiple consecutive masked/noised tokens:

**Corollary 2** (Linear growth over a missing block)**.** *If $L$ successive tokens are masked/noised ($g \leq \eta_{noise}$),*

$$\mathrm{Var}[h_{t+L}] \leq \mathrm{Var}[h_t] + L\, \eta_{noise}^2\, \sigma_n^2.$$

**Proposition 1** (Mixed-regime robustness)**.** *Let a sequence of length $T$ contain $N_{\mathrm{clean}}$ clean tokens and $N_{\mathrm{mask}}$ masked/noised tokens ($N_{\mathrm{clean}} + N_{\mathrm{mask}} = T$). Then*

$$\mathrm{Var}[h_T] \leq (1 - \eta_{clean})^{2N_{\mathrm{clean}}}\, \mathrm{Var}[h_0] + \underbrace{\frac{1 - (1 - \eta_{clean})^{2N_{\mathrm{clean}}}}{1 - (1 - \eta_{clean})^2}\, \sigma_n^2}_{\text{noise during clean steps}} + N_{\mathrm{mask}}\, \eta_{noise}^2\, \sigma_n^2. \quad (15)$$

*Hence every clean observation exponentially \*rescales\* the accumulated error, while masked/noised observations can increase it only linearly, at a rate controlled by $\eta_{noise}^2$.*

*Sketch.* Apply Lemma 1 on each clean step and Lemma 2 on each masked/noised step; telescope the products and sums to obtain equation 15. □

*Remark* 1 (Interpretation). Empirically, robustness improves when *(i)* the model rarely assigns gates below $\eta_{clean}$ on normal data, maximising the contraction factor, and *(ii)* it pushes gates close to zero on heavily corrupted inputs, minimising $\eta_{noise}$. The bound in Proposition 1 formalises this trade-off. In our experiment, we are using the masked/noised tokens, where $m_t \in \{0, 1\}$, for the proof, we give the example as:

*Example* 1 (Bernoulli mask). If $m_t \sim \text{Bernoulli}(p)$ then $\mu = p$ and $\sigma_m^2 = p(1-p)$, hence $\sigma_n^2 = p(1-p)S^2 \leq \frac{1}{4}S^2$, which recovers the specialised bound used in the original draft.

## 4.6 Training objective

We follow a variational formulation: the encoder defines a distribution over latent initial states, and the decoder produces a reconstruction of the observed sequence and a forecast of the future sequence. The generative loss includes a reconstruction term (mean-squared error over observed points) and a KL divergence regularizer between the approximate posterior and a standard normal prior. For classification, we attach a small MLP classifier to the aggregated latent state $\hat{z}_{\text{init}}$ and use binary cross-entropy for in-hospital mortality prediction.

**Joint optimization** The total loss is a weighted sum $\mathcal{L} = \mathcal{L}_{\text{gen}} + \lambda_{\text{cls}}\mathcal{L}_{\text{cls}}$, and we optimize all components (encoder, latent solver, decoder, classifier) end-to-end using Adam. This ensures that the latent representations are shaped simultaneously by generative and discriminative objectives.

## 5 Experiments

### 5.1 Experiment Setup

We conduct forecasting and classification experiments on three benchmark datasets, with an 80/10/10 split for training, validation, and testing. Following prior works Rubanova et al. (2019b); Chen et al. (2018a); Wen et al. (2023), all results are averaged over five runs with different seeds. For forecasting, the first 24 hours of patient data are used to predict the next 24, evaluated by mean squared error (MSE). For classification, in-hospital mortality is predicted from the first 24 hours, with performance measured by AUROC and AUPRC due to class imbalance. Model efficiency is reported via T-epoch, the training time per epoch Biloš et al. (2021); Li et al. (2020); Shukla & Marlin (2020), using a single NVIDIA Tesla V100 GPU.

### 5.2 Datasets

We evaluate our model on three real-world EHR datasets: MIMIC-IV (Johnson et al., 2023), PhysioNet 2012 (Goldberger et al., 2000), and eICU (Pollard et al., 2018), all consisting of multivariate, irregularly sampled ICU time series with varying sparsity and sequence lengths (Table 4). MIMIC-IV (2008–2019) includes 26,070 ICU stays with 96 variables over the first 48 hours, exhibiting extreme sparsity (missing rate $\approx 98\%$). PhysioNet 2012 provides 3,989 admissions with 37 features for mortality prediction, showing moderate sparsity. eICU (2014–2015) covers 12,312 admissions across 200+ hospitals with 14 features, and is the least sparse with relatively regular sampling. All three datasets used in our experiments are general ICU cohorts and include a wide spectrum of diagnoses; none of them are restricted to sepsis. (See Appendix A.6 for baseline details, Appendix A.7 for IVP-VAE comparisons, and Appendix A.5 for data information.)

### 5.3 Experimental Results and Analysis

Table 1 reports forecasting and classification results across three benchmarks, where Mamba-IVP consistently achieves state-of-the-art performance. Here, "$\pm$" denotes the standard deviation across five independent runs, and bold indicates the best result or results that are statistically indistinguishable from the best ($p \geq 0.05$, paired two-tailed $t$-test, $n = 5$). All values are mean $\pm$ standard deviation across five independent runs. On MIMIC-IV, it obtains the lowest MSE (0.697$\pm$0.015) and highest AUROC/AUPRC (83.2$\pm$0.5/43.8$\pm$1.5), outperforming GRU-$\Delta_t$ (0.730), GRU-D (0.736), mTAN (0.715), and IVP-VAE (AUROC 80.5, MSE 0.727). On PhysioNet 2012, it achieves MSE = 0.544$\pm$0.0034, AUROC = 79.9$\pm$3.0, and AUPRC = 39.6$\pm$2.2, surpassing GRU variants (AUROC 72.0, AUPRC 29.0), Raindrop (75.3), and IVP-VAE (77.1). On eICU, it again leads with MSE

Table 1: Forecasting and classification performance across datasets

| | MIMIC-IV (Johnson et al., 2023) | | | PhysioNet 2012 (Goldberger et al., 2000) | | | eICU (Pollard et al., 2018) | | |
|---|---|---|---|---|---|---|---|---|---|
| | MSE | AUROC | AUPRC | MSE | AUROC | AUPRC | MSE | AUROC | AUPRC |
| GRU-$\Delta_t$ (Che et al., 2018a) | 0.730±0.014 | 80.9±0.6 | 42.0±2.0 | 0.587±0.055 | 72.0±4.4 | 29.0±4.5 | 0.583±0.009 | 76.1±1.4 | 42.8±2.1 |
| GRU-D (Che et al., 2018a) | 0.736±0.005 | 78.6±0.9 | 41.9±1.3 | 0.588±0.060 | 76.2±3.2 | 32.9±4.3 | 0.578±0.007 | 79.6±1.5 | **47.7±2.4** |
| mTAN (Che et al., 2018a) | 0.715±0.011 | 76.6±0.6 | 37.9±2.4 | 0.588±0.050 | 76.2±2.2 | 33.8±5.5 | 0.582±0.010 | 76.9±2.4 | 45.1±3.2 |
| Raindrop (Zhang et al., 2022) | - | 77.1±1.4 | 36.8±2.8 | - | 75.3±2.3 | 30.9±3.9 | - | 76.6±2.1 | 45.1±2.7 |
| GOB (Brouwer et al., 2019) | 0.809±0.014 | - | - | 0.619±0.029 | - | - | 0.664±0.012 | - | - |
| CRU (Schirmer et al., 2022) | 0.946±0.016 | - | - | 0.688±0.032 | - | - | 0.820±0.044 | - | - |
| IVP-VAE (Xiao et al., 2024a) | 0.727±0.013 | 80.5±0.5 | 42.7±1.4 | 0.567±0.038 | 77.1±3.0 | 36.2±5.3 | 0.581±0.009 | 78.6±1.7 | 47.2±2.2 |
| **Mamba-IVP** | **0.697±0.015** | **83.2±0.5** | **43.8±1.5** | **0.544±0.0034** | **79.9±3.0** | **39.6±2.2** | **0.564±0.01** | **81.2±1.0** | 47.6±2.4 |

= 0.564±0.01, AUROC = 81.2±0.1, and AUPRC = 47.6±2.4, outperforming GRU-D (79.6/47.7). These results highlight Mamba-IVP's robustness across datasets of varying sparsity and irregularity.

The consistent improvements stem from two architectural strengths: the mask-aware Mamba encoder, which incorporates missingness patterns into structured state–space blocks to better capture irregular observations, and the Mamba decoder, which effectively models long-term dependencies for accurate forecasting. Together, these components enable Mamba-IVP to achieve lower errors and higher AUROC/AUPRC than prior methods, demonstrating strong generalization and robustness in clinical time series modeling.

## 5.4 ABLATION STUDY

Table 2: Ablation Study on MIMIC-IV, PhysioNet 2012 and eICU

| Setting | MIMIC-IV (Johnson et al., 2023) | | | PhysioNet 2012 (Goldberger et al., 2000) | | | eICU (Pollard et al., 2018) | | |
|---|---|---|---|---|---|---|---|---|---|
| | MSE | AUROC | AUPRC | MSE | AUROC | AUPRC | MSE | AUROC | AUPRC |
| w/o MADAE + MHD | 0.747±0.013 | 80.5±0.5 | 42.7±1.4 | 0.582±0.038 | 77.1±3.0 | 36.2±5.3 | 0.592±0.009 | 78.6±1.7 | 47.7±2.4 |
| w/o MHD | 0.724±0.01 | 81.2±0.6 | 43.7±1.5 | 0.554±0.003 | 78.1±3.0 | 38.2±4.3 | 0.575±0.01 | 79.8±1.8 | 47.5±2.2 |
| w/o MADAE | 0.727±0.01 | 82.3±0.5 | 43.6±1.5 | 0.559±0.003 | 79.8±3.0 | **40.3±5.2** | 0.572±0.01 | 80.8±2.0 | **48.2±2.6** |
| Mamba-IVP | **0.697±0.015** | **83.2±0.5** | **43.8±1.5** | **0.544±0.0034** | **79.9±3.0** | 39.6±2.2 | **0.564±0.01** | **81.2±1.0** | 47.6±2.4 |

We conduct ablation experiments on three clinical time-series datasets, MIMIC-IV, PhysioNet 2012, and eICU, to investigate the effectiveness of the proposed components, including the MADAE and the MHD. As shown in the results, removing either MADAE or MHD leads to noticeable degradation across all metrics, confirming that both components contribute to the model's performance.

It is worth noting that the AUROC and AUPRC metrics exhibit relatively large standard errors, particularly on the PhysioNet 2012 and eICU datasets. This behavior is attributed to the inherent data characteristics, such as extreme class imbalance and irregular sampling patterns, that amplify the variability in binary classification performance. In highly imbalanced settings, minor variations in model behavior or sample distributions can cause significant fluctuations in AUROC and AUPRC. Importantly, this phenomenon is not unique to our model. As illustrated in Table 1, all baseline models, including GRU-$\Delta_t$, GRU-D, mTAN, Raindrop, GOB, CRU, and IVP-VAE, also exhibit substantial variance in AUROC and AUPRC. This reinforces the conclusion that such fluctuations stem from the nature of the data, not from model instability. The visualization results of our ablation studies are provided in Appendix A.11.

## 5.5 ROBUSTNESS STUDY

Tables 3(a) and (b) evaluate robustness under increasing noise levels and block-wise missingness, respectively. We report both MAE and MSE as complementary metrics. These results align with our theoretical analysis in Section 4.5, which shows that Mamba gates contract variance exponentially on clean tokens while allowing only linear growth under noisy or block-masked inputs. This variance-bound property yields strong denoising ability. Across both settings, Mamba-IVP consistently achieves lower MAE and MSE than IVP-VAE, demonstrating greater resilience to perturbations and block-wise missingness. Under noise corruption (Table 3a), Mamba-IVP maintains errors ranging from MAE 0.563–0.571 and MSE 0.706–0.724, while IVP-VAE degrades markedly (MSE > 1.2). Under block-wise missingness of 2–12 hours (Table 3b), Mamba-IVP sustains low error

Table 3: Validation robustness of Mamba-IVP and IVP-VAE under (a) different noise levels and (b) different levels of temporal missingness.

(a) Noise levels

| Noise | Mamba-IVP | | IVP-VAE | |
|---|---|---|---|---|
| | MAE ↓ | MSE ↓ | MAE ↓ | MSE ↓ |
| 0.1 | 0.5630 | 0.7081 | 0.6119 | 1.2131 |
| 0.2 | 0.5631 | 0.7172 | 0.6037 | 1.2278 |
| 0.3 | 0.5647 | 0.7059 | 0.6025 | 1.2334 |
| 0.4 | 0.5718 | 0.7237 | 0.6082 | 1.2447 |
| 0.5 | 0.5657 | 0.7090 | 0.6129 | 1.2563 |

(b) Block-wise missingness

| Missing | Mamba-IVP | | IVP-VAE | |
|---|---|---|---|---|
| | MAE ↓ | MSE ↓ | MAE ↓ | MSE ↓ |
| 2h | 0.4938 | 0.5628 | 0.5102 | 0.5912 |
| 5h | 0.4928 | 0.5616 | 0.5017 | 0.5907 |
| 7h | 0.5003 | 0.5859 | 0.5109 | 0.6163 |
| 10h | 0.4918 | 0.5937 | 0.5152 | 0.6402 |
| 12h | 0.5035 | 0.6090 | 0.5349 | 0.6907 |

(e.g., MAE $0.4928$, MSE $0.5616$ at a 5-hour gap) with only modest increases, whereas IVP-VAE shows consistently higher errors.

These results empirically confirm our variance-bound theory: clean tokens rapidly stabilise hidden states through contraction, while noisy or block-wise missingness only induce controlled linear error growth. The synergy of mask-aware encoding and hybrid decoding thus enables Mamba-IVP to remain reliable under noisy and incomplete conditions, a critical property for deployment in real-world ICUs where multi-hour gaps and sensor outages are routine.

## 5.6 EFFICIENCY ANALYSIS

ODE- and RNN-based models face computational bottlenecks: ODEs require costly iterative solvers, while RNNs are inherently sequential, both hindering real-time ICU use. To overcome this, Mamba-IVP employs parallelizable state–space dynamics that cut training and inference costs. Efficiency analysis shows it achieves the best trade-off between accuracy, latency, and cost, delivering the lowest error with far less computation than all baselines. Full comparisons are given in Appendix A.12. All additional diffusion results, imputation evaluations, two-stage baselines, and full parameter and efficiency comparisons are provided in Appendix A.8, Appendix A.9, and Appendix A.10, respectively.

## 6 CONCLUSION

In this work, we propose Mamba-IVP, a novel framework for continuous-time modeling that integrates a Mask-Aware Dual-Mamba Encoder (MADME) to address block-wise missingness with a Mamba-Hybrid Decoder (MHD) to handle sensor noise from aging devices. By leveraging state–space inspired sequence modeling, our approach robustly handles irregular sampling, block-wise missingness, and device-induced noise. Theoretical analysis shows that Mamba gates contract variance on clean inputs while controlling error growth under missingness and noise, and our empirical results across multiple clinical datasets confirm this property, with Mamba-IVP consistently outperforming IVP-VAE and other strong baselines. Beyond predictive accuracy, we introduced an efficiency–accuracy trade-off metric and demonstrated that Mamba-IVP achieves state-of-the-art performance while reducing both forward time and epoch time, underscoring its practicality for real-world deployment in resource-constrained clinical environments.

## 7 REPRODUCIBILITY STATEMENT

To ensure reproducibility, we provide our source code, detailed training configurations, and pre-processed datasets as supplementary material. The implementation includes all hyperparameter settings, random seeds, and evaluation scripts, enabling independent verification of both forecasting and classification results.

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

# A APPENDIX

## A.1 SELECTIVE STATE–SPACE MODEL

The Selective State–Space Model (Mamba) was recently proposed by Gu & Dao (2024) as a sequence modeling framework based on continuous-time linear dynamical systems. Its formulation starts from the continuous-time, linear time-invariant (LTI) state–space equations:

$$\frac{d\mathbf{h}(\tau)}{d\tau} = \mathbf{A}\mathbf{h}(\tau) + \mathbf{B}\mathbf{x}(\tau), \qquad \mathbf{y}(\tau) = \mathbf{C}\mathbf{h}(\tau), \qquad (16)$$

where $\mathbf{x}(\tau)$, $\mathbf{h}(\tau)$, and $\mathbf{y}(\tau)$ denote input, hidden state, and output at time $\tau$, and $\mathbf{A}, \mathbf{B}, \mathbf{C}$ are learnable matrices. For discretely sampled data with interval $\Delta$, Zero-Order Hold (ZOH) discretization yields:

$$\mathbf{h}_{t+1} = \exp(\mathbf{A}\Delta)\mathbf{h}_t + \widetilde{\mathbf{B}}\mathbf{x}_t, \quad \widetilde{\mathbf{A}} = \exp(\Delta\mathbf{A}), \quad \widetilde{\mathbf{B}} = \mathbf{A}^{-1}(\exp(\Delta\mathbf{A}) - \mathbf{I})\mathbf{B} \approx \Delta\mathbf{B}. \quad (17)$$

A key feature of Mamba is its *selective parameterization*: instead of fixing $\mathbf{A}, \mathbf{B}, \mathbf{C}$, they are dynamically generated from the current input,

$$\mathbf{A}_\tau = f_A(\mathbf{x}(\tau)), \quad \mathbf{B}_\tau = f_B(\mathbf{x}(\tau)), \quad \mathbf{C}_\tau = f_C(\mathbf{x}(\tau)), \qquad (18)$$

allowing adaptive dynamics while retaining the efficiency of linear state–space systems.

## A.2 ALGORITHM

---

**Algorithm 1:** Mamba-IVP with Mask-Aware Dual-Mamba Encoder and Mamba-Hybrid Decoder

---

1 **Input:** $\mathbf{X} \in \mathbb{R}^{L \times D}$ (observed sequence), $\mathbf{M} \in \{0,1\}^{L \times D}$ (mask), $t_{\text{in}}, t_{\text{out}} \in \mathbb{R}^L$ (input/output timestamps)

2 **Output:** $\hat{\mathbf{X}} \in \mathbb{R}^{L \times D}$ (reconstructed sequence)

  1: $\tilde{\mathbf{X}} \leftarrow \text{Concat}(\mathbf{X}, \mathbf{M})$

  2: $\mathbf{Z} \leftarrow \text{MADAE}(\tilde{\mathbf{X}})$

      $\mathbf{H}^0 \leftarrow \text{LayerNorm}(\tilde{\mathbf{X}}) \cdot \mathbf{W}_{\text{proj}} + \mathbf{b}_{\text{proj}}$

      $\mathbf{H}^1 \leftarrow \mathbf{H}^0 + \text{Mamba}_1(\mathbf{H}^0)$

      $\mathbf{H}^2 \leftarrow \mathbf{H}^1 + \text{Mamba}_2(\text{Dropout}(\mathbf{H}^1))$

      $\mathbf{Z} \leftarrow \text{LayerNorm}(\mathbf{H}^2) \cdot \mathbf{W}_{\text{out}} + \mathbf{b}_{\text{out}}$

  3: $\mathbf{Z}^{\leftarrow}(t) \leftarrow \text{IVPSolver}()$

  4: $\hat{\mathbf{z}}_{\text{init}} \leftarrow \text{Aggregate}(\mathbf{Z}^{\leftarrow}(t))$

  5: $\mathbf{Z}^{\rightarrow}(t) \leftarrow \text{IVPSolver}()$

  6: $\hat{\mathbf{X}} \leftarrow \text{MHD}(\mathbf{Z}^{\rightarrow}(t))$

      $\mathbf{H} \leftarrow \mathbf{Z}^{\rightarrow}(t) \cdot \mathbf{W}_{\text{in}}^{\top} + \mathbf{b}_{\text{in}}$

      $\hat{\mathbf{H}} \leftarrow \text{LayerNorm}(\text{Mamba}(\mathbf{H} + \mathbf{H}))$

      $\hat{\mathbf{X}} \leftarrow \text{ReLU}(\hat{\mathbf{H}} \cdot \mathbf{W}_1^{\top} + \mathbf{b}_1) \cdot \mathbf{W}_2^{\top} + \mathbf{b}_2$

  7: **return** $\hat{\mathbf{X}}$

---

## A.3 DETAILED PROOF FOR LEMMA 1

*Detailed proof for Lemma 1.* Start from the selective state–space update equation 14 and mean-centre all variables. Define

$$\tilde{h}_t := h_t - \mathbb{E}[h_t], \qquad \tilde{n}_t := n_t = x_t - \mathbb{E}[x_t] = (m_t - \mu)\, s_t,$$

where $x_t = m_t s_t$ and the mask sequence $\{m_t\}_{t\geq 0}$ is i.i.d. and independent of the past. Here

$$\mu := \mathbb{E}[m_t], \qquad \sigma_m^2 := \mathrm{Var}[m_t] < \infty, \qquad \sigma_n^2 := \sigma_m^2 S^2, \text{ with } S = \sup_{t\geq 0} |s_t|.$$

Thus $\mathbb{E}[\tilde{n}_t] = 0$ and $\mathrm{Var}[\tilde{n}_t] \leq \sigma_n^2$ uniformly in $t$.

Subtracting expectations from equation 14 gives

$$\tilde{h}_t = (1 - g_t)\, \tilde{h}_{t-1} + g_t\, \tilde{n}_t.$$

Because $n_t$ is independent of $\tilde{h}_{t-1}$ (mask process independent of past states), the mixed expectation vanishes: $\mathbb{E}\big[\tilde{h}_{t-1} n_t\big] = 0$. Hence

$$
\begin{aligned}
\mathrm{Var}[h_t] &= \mathbb{E}\big[\tilde{h}_t^2\big] \\
&= (1 - g_t)^2\, \mathbb{E}\big[\tilde{h}_{t-1}^2\big] + g_t^2\, \mathbb{E}\big[n_t^2\big] \\
&= (1 - g_t)^2\, \mathrm{Var}[h_{t-1}] + g_t^2\, \mathrm{Var}[n_t].
\end{aligned}
\tag{19}
$$

Now, under the assumption of bounded signal and finite variance of $m_t$

$$\mathrm{Var}[n_t] \leq \sigma_n^2 \quad \text{for all } t.$$

When the current token is *clean* we have the gate lower bound $g_t \geq \eta_{clean}$ for some constant $0 < \eta_{clean} < 1$. Consequently

$$(1 - g_t)^2 \leq (1 - \eta_{clean})^2, \qquad g_t^2 \leq 1.$$

Insert these bounds into equation 19:

$$\mathrm{Var}[h_t] \leq (1 - \eta_{clean})^2\, \mathrm{Var}[h_{t-1}] + 1 \cdot \sigma_n^2,$$

which is exactly the desired inequality,

$$\boxed{\mathrm{Var}[h_t] \leq (1 - \eta_{clean})^2\, \mathrm{Var}[h_{t-1}] + \sigma_n^2}.$$

$\square$

## A.4 DETAIL PROOF OF LEMMA 2

*Proof.* The proof of Lemma 2 is done in the same way of Lemma 1. Firstly, recall the update rule

$$h_t = (1 - g_t)\, h_{t-1} + g_t\, x_t, \qquad \text{where} \qquad x_t = m_t\, s_t.$$

Then, define the centred (zero-mean) versions

$$\tilde{h}_{t-1} := h_{t-1} - \mathbb{E}[h_{t-1}], \qquad n_t := x_t - \mathbb{E}[x_t] = (m_t - \mu)\, s_t.$$

with $\mathbb{E}[n_t] = 0$, we have

$$h_t - \mathbb{E}[h_t] = (1 - g_t)\, \tilde{h}_{t-1} + g_t\, n_t =: \tilde{(h_t)}.$$

Using independence of $m_t$ from $\{h_\tau\}_{\tau < t}$ (hence of $n_t$ from $\tilde{h}_{t-1}$), the cross-term vanishes:

$$\mathbb{E}\big[\tilde{h}_{t-1}\, n_t\big] = \mathbb{E}[\tilde{h}_{t-1}]\, \mathbb{E}[n_t] = 0.$$

Therefore

$$
\begin{aligned}
\mathrm{Var}[h_t] &= \mathbb{E}\big[\tilde{h}_t^2\big] \\
&= (1 - g_t)^2\, \mathbb{E}\big[\tilde{h}_{t-1}^2\big] \ + \ g_t^2\, \mathbb{E}\big[n_t^2\big] \\
&= (1 - g_t)^2\, \mathrm{Var}[h_{t-1}] \ + \ g_t^2\, \mathrm{Var}[n_t].
\end{aligned}
\tag{20}
$$

The uniform noise bound is constructed as $\mathrm{Var}[n_t] \le \sigma_n^2$ holds for every $t$.

In the gate bounds under masking token we assume $g_t \le \eta_{noise}$ with some fixed constant $0 < \eta_{noise} < 1$. Consequently,

$$
(1 - g_t)^2 \ \le \ 1, \qquad g_t^2 \ \le \ \eta_{noise}^2.
$$

Combining these inequalities and inserting them into equation 20:

$$
\mathrm{Var}[h_t] \ \le \ 1 \cdot \mathrm{Var}[h_{t-1}] \ + \ \eta_{noise}^2\, \sigma_n^2,
$$

which is exactly the claim of Lemma 2:

$$
\boxed{\mathrm{Var}[h_t] \ \le \ \mathrm{Var}[h_{t-1}] \ + \ \eta_{noise}^2\, \sigma_n^2}.
$$

$\square$

## A.5 DATASET INFORMATION

Table 4: Key statistics of the three EHR datasets used in experiments

| Dataset | Samples | Variables | Missing Rate | Avg. Length | Mortality Rate |
|---|---|---|---|---|---|
| MIMIC-IV | 26,070 | 96 | 97.95% | 173.4 | 13.39% |
| PhysioNet 2012 | 3,989 | 37 | 84.34% | 75.0 | 13.89% |
| eICU | 12,312 | 14 | 65.25% | 114.55 | 17.61% |

## A.6 BASELINES

We compare our model against several representative baselines for forecasting and classification of multivariate irregular time series.

- **GRU-$\Delta_t$.** This baseline concatenates observed values with masking indicators and time intervals $\Delta_t$ to handle missingness in time series data (Che et al., 2018a).

- **GRU-D.** GRU-D incorporates missing patterns by employing a gating mechanism and a learnable decay function applied to both input values and hidden states (Che et al., 2018a).

- **mTAN.** The Multi-Time Attention Network leverages temporal attention mechanisms and time embeddings to capture dependencies across irregular time points (Narayan Shukla & Marlin, 2021).

- **GRU-ODE-Bayes.** This method couples continuous-time dynamics modeled by ODEs with discrete-time Bayesian update steps to form a hybrid sequential model (Brouwer et al., 2019).

- **CRU.** The Continuous Recurrent Unit constructs continuous-time recurrent cells based on linear stochastic differential equations and Kalman filtering, providing a probabilistic framework for irregular sequences (Schirmer et al., 2022).

- **Raindrop.** Raindrop models multivariate dependencies using a learned graph structure, with temporal irregularities captured through graph attention mechanisms (Zhang et al., 2022).

- **Latent-ODE.** Latent-ODE employs an ODE-RNN encoder and a neural ODE decoder within a variational autoencoder (VAE) framework to learn latent dynamics in continuous time (Rubanova et al., 2019a).

- **Latent-Flow.** Latent-Flow improves upon Latent-ODE by replacing the ODE solver with a more efficient invertible neural flow, while keeping the VAE architecture (Rubanova et al., 2019a).

- **IVP-VAE.** Eliminates recurrent structures by solving initial value problems (IVPs) in parallel, and shares one IVP solver between encoder and decoder by leveraging its invertibility. It achieves faster training and improved efficiency in modeling irregularly sampled time series (Xiao et al., 2024a).

Our model is designed based on the VAE + IVP pattern, so IVP-VAE is the main benchmark of our model. Moreover, we evaluate our mamba-IVP using two types of ordinary differential equation solvers, namely those based on ODE and flow.

## A.7  THE RESULTS COMPARED WITH BENCHMARKS

**Mamba-IVP vs. IVP-VAE Comparative Analysis**  As shown in Table 5, our proposed Mamba-IVP consistently achieves superior performance across datasets and tasks. On the PhysioNet 2012 dataset, Mamba-IVP yields the lowest MSE of 0.537, outperforming IVP-VAE (0.563). For classification, it achieves an AUROC of 0.799 and an AUPRC of 0.362, both higher than IVP-VAE (0.770, 0.359) and also surpassing other baselines such as mTAN (0.762, 0.338). Similar trends are observed on MIMIC-IV, where Mamba-IVP achieves an MSE of 0.690 (vs. 0.724 for IVP-VAE), an AUROC of 0.822 (vs. 0.802), and an AUPRC of 0.432 (vs. 0.422). Even on the eICU dataset, which presents greater temporal irregularity, Mamba-IVP maintains strong generalizability with an AUROC of 0.815 (vs. 0.786) and a lower MSE of 0.578 (vs. 0.596).

These improvements stem from the architectural innovations of our model. Unlike traditional IVP-VAEs that rely on MLP or GRU-based modules, our MambaEmbedding layer employs state–space sequence modeling, which efficiently captures long-range temporal dependencies. Additionally, it incorporates observation masks by concatenating them with raw inputs, ensuring robust encoding under missingness. In the decoding stage, we adopt the Mamba-Hybrid Decoder, which replaces the conventional MLP decoder with a structured Mamba block followed by a lightweight feedforward head, enhancing temporal extrapolation and denoising capacity.

Notably, these gains do not come at the expense of efficiency. While Mamba-IVP introduces a modestly larger parameter count (e.g., 541K on PhysioNet classification vs. 174K for IVP-VAE), it achieves faster runtime. For instance, on MIMIC-IV forecasting, Mamba-IVP reduces per-forward inference time (T-forward) from 0.106s to 0.032s and per-epoch training time (T-epoch) from 155.4s to 72.4s. This acceleration arises from Mamba's ability to parallelize computations across time steps, avoiding the recursive bottlenecks inherent in ODE solvers and GRUs.

In summary, Mamba-IVP simultaneously achieves lower prediction error, higher classification accuracy, and faster training/inference, demonstrating that thoughtful architectural design—specifically, the integration of Mamba-based encoder and decoder modules—yields both performance and efficiency gains in continuous-time modeling.

## A.8  COMPARISON BETWEEN MAMBA-IVP AND EXISTING IMPUTATION METHODS

Table 6 reports the imputation accuracy on the PhysioNet 2012 dataset under three controlled missingness settings—30%, 50%, and 70%. PhysioNet 2012 is a real-world ICU multivariate physiological time-series benchmark that naturally exhibits irregular sampling and structured missingness, making it well suited for evaluating models under challenging incomplete-data conditions. We compare four representative categories of imputation approaches: (1) classical non-neural methods (MissForest); (2) neural network–based imputers (SAITS); (3) diffusion-based generative models (CSDI); and (4) continuous-time generative models, including the baseline IVP-VAE and our proposed Mamba-IVP. These categories cover the dominant paradigms in time-series imputation—from traditional statistics to modern attention-based imputers and diffusion models—allowing a comprehensive evaluation under increasing sparsity. As shown in Table 6, the superiority of Mamba-IVP is consistent across all missingness ratios and can be quantified directly. At 30% missingness, Mamba-IVP achieves an RMSE of 0.76, outperforming SAITS (0.97) by 0.21, CSDI (0.90) by 0.14, and the baseline IVP-VAE (0.79) by 0.03. When the missingness increases to 50%, Mamba-IVP maintains strong robustness with an RMSE of 0.78, improving over SAITS (1.00) by 0.22 and CSDI (0.94)

Table 5: Benchmark results across datasets for ODE/Flow models with Mamba-IVP and IVP-VAE

| Dataset | Task | Metric | ODE-Mamba-IVP (Ours) | ODE-IVP-VAE | Flow-Mamba-IVP (Ours) | Flow-IVP-VAE |
|---|---|---|---|---|---|---|
| MIMIC-IV | Classification | AUROC | **0.822** | 0.802 | **0.832** | 0.805 |
| | | AUPRC | **0.432** | 0.422 | **0.428** | 0.427 |
| | | T-forward | **0.032** | 0.066 | **0.009** | 0.017 |
| | | T-epoch | **984.4** | 1478.8 | **949.1** | 1445.8 |
| | | # Epochs | **10** | 12.6 | 15 | **10.8** |
| | | # Parameters | 688,313 | **209,677** | 737,574 | **325,017** |
| | Forecasting | MSE | **0.690** | 0.724 | **0.697** | 0.727 |
| | | T-forward | **0.032** | 0.106 | **0.014** | 0.025 |
| | | T-epoch | **72.4** | 155.4 | **51.6** | 81.5 |
| | | # Epochs | **25** | 31.8 | **28** | 35.6 |
| | | # Parameters | 548,888 | **112,776** | 663,312 | **228,116** |
| PhysioNet 2012 | Classification | AUROC | **0.799** | 0.770 | **0.797** | 0.771 |
| | | AUPRC | **0.362** | 0.359 | **0.363** | 0.362 |
| | | T-forward | **0.015** | 0.031 | **0.005** | 0.009 |
| | | T-epoch | **23.7** | 35.6 | **21.4** | 32.6 |
| | | # Epochs | **12** | 19.6 | 28 | **19.4** |
| | | # Parameters | 541,766 | **174,218** | 657,106 | **289,558** |
| | Forecasting | MSE | **0.537** | 0.563 | **0.544** | 0.567 |
| | | T-forward | **0.022** | 0.072 | **0.007** | 0.012 |
| | | T-epoch | **9.4** | 20.2 | **5.2** | 8.2 |
| | | # Epochs | **31** | 54.4 | **30** | 68.0 |
| | | # Parameters | 444,865 | **77,317** | 560,205 | **192,657** |
| eICU | Classification | AUROC | **0.815** | 0.786 | **0.812** | 0.786 |
| | | AUPRC | **0.472** | 0.468 | **0.474** | 0.472 |
| | | T-forward | **0.015** | 0.033 | **0.005** | 0.009 |
| | | T-epoch | **228.0** | 342.5 | **209.6** | 319.4 |
| | | # Epochs | **14** | 16.0 | 30 | **23.0** |
| | | # Parameters | 498,780 | **160,395** | 625,736 | **275,735** |
| | Forecasting | MSE | **0.578** | 0.596 | **0.564** | 0.581 |
| | | T-forward | **0.022** | 0.081 | **0.007** | 0.012 |
| | | T-epoch | **38.2** | 77.7 | **20.3** | 28.2 |
| | | # Epochs | **34** | 60.2 | **36** | 78.2 |
| | | # Parameters | 498,780 | **160,395** | 625,736 | **275,735** |

Table 6: Imputation performance (RMSE) on PhysioNet 2012.

| Method | 30% Missing | 50% Missing | 70% Missing |
|---|---|---|---|
| MissForest | 1.34 | 1.42 | 1.48 |
| SAITS | 0.97 | 1.00 | 1.04 |
| CSDI | 0.90 | 0.94 | 0.99 |
| IVP-VAE | 0.79 | 0.83 | 0.88 |
| **Mamba-IVP** | **0.76** | **0.78** | **0.82** |

by 0.16. Under the most challenging 70% missing scenario, the gap becomes even clearer: Mamba-IVP achieves 0.82, whereas SAITS and CSDI degrade to 1.04 and 0.99, respectively. Even compared with the continuous-time IVP-VAE (0.88), Mamba-IVP reduces the error by 0.06. These concrete numerical gains demonstrate that Mamba-IVP not only improves average accuracy but also preserves stability as sparsity increases, particularly outperforming diffusion-based CSDI under high missingness conditions.

## A.9 DOWNSTREAM EVALUATION AFTER IMPUTATION: TWO-STAGE VS. END-TO-END

Table 7: Two-stage vs. end-to-end performance on PhysioNet 2012 (50 Epochs).

| Method | Type | MSE ↓ | AUROC ↑ | Total Time (s) ↓ |
|---|---|---|---|---|
| MissForest → GRU | Two-stage | 0.792 | 0.55 | 414 |
| SAITS → GRU | Two-stage | 0.743 | 0.64 | 6186 |
| CSDI → MLP | Two-stage | 0.758 | 0.62 | 1820 |
| IVP-VAE | End-to-end | 0.567 | 0.77 | 1630 |
| **Mamba-IVP** | End-to-end | **0.544** | **0.80** | **1070** |

Table 7 reports the downstream results on forecasting (MSE), classification (AUROC), and over-all computation cost (Total Time). While the previous subsection focused on comparing imputation accuracy, this experiment evaluates whether different imputation strategies can actually support stronger downstream task performance. The three two-stage pipelines show clear limitations: MissForest→GRU produces the weakest results with an MSE of 0.792 and an AUROC of 0.55. SAITS→GRU improves the classification score (0.64) but still yields a relatively large forecasting error (0.743) and incurs a massive runtime of 6186 seconds. CSDI→MLP obtains moderate performance (0.758 MSE, 0.62 AUROC) but remains inferior to continuous-time models. These results indicate that even strong imputers may not preserve temporal consistency after reconstruction, leading to error propagation when the forecasting or classification model is applied.

End-to-end approaches, in contrast, jointly optimize representations for both forecasting and classification directly from the incomplete time series, yielding substantially stronger performance. IVP-VAE already surpasses all two-stage baselines with an MSE of 0.567 and an AUROC of 0.77 while reducing total computation to 1630 seconds. Our proposed **Mamba-IVP** further improves every metric, achieving the best forecasting accuracy (MSE **0.544**), the highest classification score (AUROC **0.80**), and the lowest compute cost among learnable models (1070 seconds). Compared with IVP-VAE, Mamba-IVP lowers the forecasting error by 0.023, increases AUROC by 0.03, and reduces total time by 34%, demonstrating that a selective state-space architecture provides both higher efficiency and more reliable downstream performance.

## A.10 COMPUTATIONAL EFFICIENCY COMPARISON WITH BASELINE MODELS

Table 8: Computational efficiency on PhysioNet 2012 (same hardware/settings).

| Method | #Params (M) | MSE ↓ | T-forward (s)↓ | T-epoch (s)↓ | Peak Memory (MB)↓ |
|---|---|---|---|---|---|
| GRU-D | 154,113 | 0.586 | 0.185 | 130.6 | 342 |
| mTAN | 348,672 | 0.592 | 0.243 | 195.4 | 546 |
| Latent-flow | 421,980 | 0.586 | 0.307 | 264.7 | 720 |
| IVP-VAE | 192,657 | 0.568 | 0.012 | 8.2 | 164 |
| **Mamba-IVP** | **560,205** | **0.542** | **0.007** | **5.2** | **245** |

Table 8 presents a controlled comparison of computational efficiency across all baseline models. All experiments are conducted on a single NVIDIA RTX 4090 GPU (24GB VRAM), and every model is trained for exactly 50 epochs without early stopping to ensure comparability of total runtime. To keep the optimization setup consistent, we use a unified batch size of 50, a learning rate of $1 \times 10^{-3}$, and the Adam optimizer for all methods. For fairness in model capacity, we also fix the hidden dimension to 64 across GRU-D, mTAN, Latent-flow, IVP-VAE, and our Mamba-IVP. This ensures that differences in forward speed, epoch time, and GPU memory usage arise from architectural characteristics rather than changes in hyperparameters.

The only parameter that cannot be strictly unified is the sequence length. Discrete-time baselines (GRU-D, mTAN, Latent-flow, IVP-VAE) require a fixed-length window and therefore use a padded sequence length of 48. In contrast, Mamba-IVP follows a continuous-time formulation and directly consumes irregular timestamps, eliminating the need for a predefined observation window. Aside from this unavoidable structural difference, all training settings are fully matched.

A notable observation in Table 8 is that **Mamba-IVP has the highest parameter count (560K)** among all compared models, yet still achieves both the **best forecasting accuracy** (MSE = 0.542) and the **fastest computation**. This result is not paradoxical: the additional parameters in Mamba-IVP are not used for deeper sequential computation, but are instead allocated to the input projection, selective state-space kernels, and output mixing modules, *all of which run fully in parallel* through the prefix-scan based state update. In contrast, RNN-like baselines allocate parameters into recurrent weights that must be applied step-by-step, increasing temporal computation cost.

Although Mamba-IVP's parallel recurrence introduces additional intermediate buffers, leading to a slightly higher peak memory usage compared with IVP-VAE (245 MB vs. 164 MB), this overhead is modest and highly favorable. The model obtains a richer dynamic representation of continuous-time trajectories while still completing each forward pass in only 0.007 seconds and each epoch in 5.2 seconds. In other words, **the larger parameter budget enhances representational power rather than computational burden**, enabling Mamba-IVP to simultaneously achieve superior accuracy and the fastest runtime among all baselines.

## A.11 VISUALIZATION OF ABLATION EXPERIMENTS

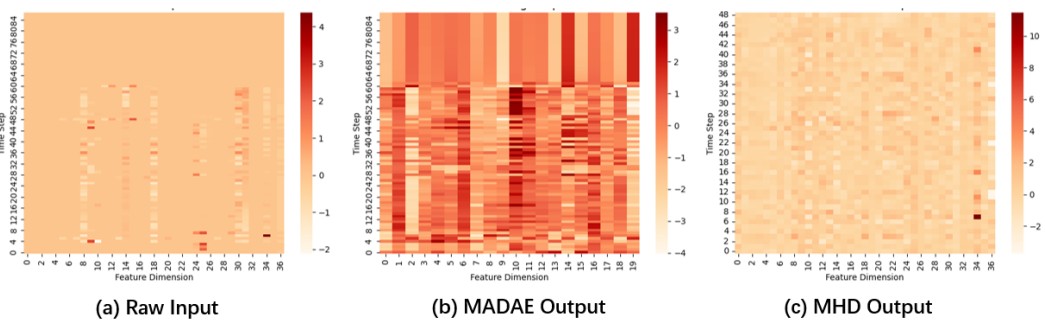

| (a) Raw Input | (b) MADAE Output | (c) MHD Output |

Figure 2: Module output heatmap

Figure 2 offers qualitative evidence of how each module progressively enhances the representation quality. Specifically, the heatmap of the (a) raw input appears highly sparse and noisy, with weak activations and irregular patterns across the time steps, particularly at the target node. After passing through the (b) MADAE, the output becomes noticeably more structured, showing clear activation bands that align with meaningful temporal segments. This reflects the module's ability to extract salient temporal dynamics from irregular and partially observed sequences. Subsequently, the decoded output from the (c) MHD further refines these representations. Compared to the raw input and embedding stages, the final heatmap exhibits smoother transitions, reduced noise, and denser, more informative temporal patterns—particularly in the central region and later time steps. This demonstrates that the decoder not only preserves the informative structure generated by the encoder but also enhances it by modeling long-range dependencies.

Taken together, these visualizations clearly highlight that each stage, embedding and decoding, contributes to denoising, pattern sharpening, and temporal abstraction. The visual progression in Figure 2 complements our quantitative gains, reinforcing the effectiveness and interpretability of the proposed architecture.

## A.12 EFFICIENCY ANALYSIS

In high-stakes domains such as clinical decision support and patient monitoring, computational efficiency is as critical as predictive accuracy. However, diffusion- and flow-based models, despite their strong performance, incur prohibitive costs due to iterative solvers and sequential updates, making them unsuitable for real-time deployment. To overcome these bottlenecks, we adopt Mamba's parallelizable state–space dynamics, which replace recursive operations with scan-based updates and enable linear-time sequence modeling. This design not only reduces inference latency and training overhead but also preserves modeling capacity. As such, the practicality of a time-series forecast-

Table 9: Efficiency-Accuracy Trade-off Comparison on PhysioNet (lower is better)

| Model | MSE | $T_{\textbf{forward}}$ (s) | $T_{\textbf{epoch}}$ (s) | Trade-off Score |
|---|---|---|---|---|
| **Mamba-IVP** | **0.544** | **0.007** | **5.2** | **0.0198** |
| IVP-VAE-Flow | 0.567 | 0.012 | 8.2 | 0.0558 |
| GRU-$\Delta_t$ | 0.587 | 0.039 | 111.3 | 2.5463 |
| GRU-D | 0.588 | 0.185 | 130.6 | 14.1708 |
| Latent-Flow | 0.584 | 0.307 | 264.7 | 47.453 |

ing model must be judged holistically, accounting for accuracy, training cost, and inference speed together.

To operationalize this, we define a unified efficiency-accuracy trade-off score:

$$\text{Trade-off Score} = \text{MSE} \times T_{\text{forward}} \times T_{\text{epoch}}, \tag{21}$$

where MSE measures prediction error, $T_{\text{forward}}$ denotes inference latency, and $T_{\text{epoch}}$ captures training speed. This multiplicative formulation penalizes models that are either inaccurate, slow to infer, or inefficient to train—factors that jointly determine deployability in clinical pipelines.

These trade-offs are visually summarized in Figure 3, which presents a bubble plot where the X-axis denotes inference latency ($T_{\text{forward}}$), the Y-axis reflects forecasting error (MSE), and the size of each bubble represents training time per epoch ($T_{\text{epoch}}$). A color gradient encodes the composite trade-off score, with darker hues indicating worse efficiency. In this visual space, ideally efficient models should reside in the lower-left region with small, bright-colored bubbles—signifying accurate, fast, and lightweight behavior. Mamba-IVP stands out in this figure as the most favorable candidate, positioned at the extreme lower-left with the smallest bubble and lightest color.

This metric exposes a stark disparity in the efficiency profile of current approaches. While several baselines attain comparable accuracy, their computational overhead severely limits real-world utility. For instance, Latent-Flow, a continuous-time generative model built on neural ODEs, achieves a respectable MSE of $0.584$, yet incurs a forward-pass latency of $0.307$ seconds and a per-epoch training cost of $264.7$ seconds. These figures reflect the inherent inefficiencies of ODE solvers: solver adaptivity introduces latency variance, and the backpropagation via adjoint sensitivity further exacerbates training cost.

In contrast, our proposed **Mamba-IVP** achieves not only the lowest error (MSE $= 0.544$), but does so with dramatically reduced compute demand. Its inference latency is over *40×* faster than Latent-Flow, and training cost is reduced by *50×*, yielding a trade-off score of $0.0198$—the lowest among all models evaluated. This efficiency is rooted in its architecture: Mamba-IVP replaces recurrent or ODE-based temporal modeling with a state–space–inspired Mamba block, enabling fully parallelized sequence updates via scan operations. Unlike GRU-based models, which suffer from inherently sequential recurrence, or ODE-based models that require costly numerical integration, Mamba-IVP exhibits linear-time scaling and constant-time inference regardless of sequence length.

Interestingly, even lightweight RNN variants such as GRU-D or GRU-$\Delta t$ exhibit inferior trade-offs. Despite having fewer parameters, their reliance on gated recurrence prevents efficient GPU utilization and contributes to elevated per-epoch training times. Their MSE scores also lag behind, suggesting a compromised balance between modeling capacity and temporal abstraction.

Ultimately, the analysis reveals that efficiency and accuracy are not necessarily at odds, when model architectures are designed with both algorithmic and hardware characteristics in mind, it is possible to achieve superior predictive fidelity without incurring training or deployment bottlenecks. In the clinical context, such properties are not merely desirable but essential.

## A.13 DISCUSSION

To further ensure that the efficiency and accuracy gains of Mamba-IVP are not simply the result of having more parameters than IVP-VAE, we perform an additional parameter-controlled ablation where we reduce the embedding and latent dimensions of Mamba-IVP so that its total parameter count closely matches that of IVP-VAE (difference within ±5%), see Table 10. Even under

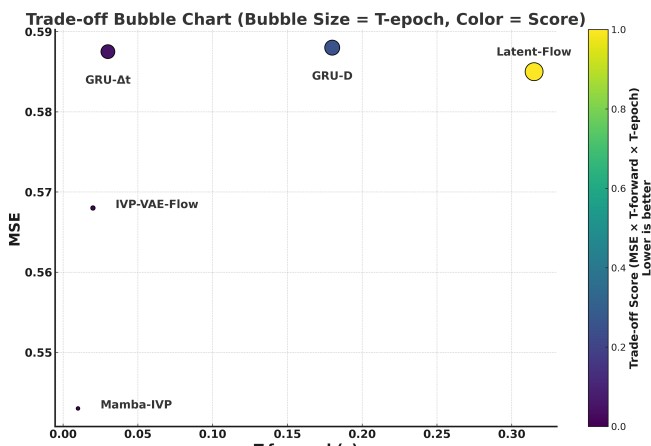

Figure 3: Trade-off comparison of models in terms of MSE, inference time, and training efficiency.

this parameter-matched configuration, Mamba-IVP continues to outperform IVP-VAE across all three datasets: forecasting MSE remains 2–3% lower on PhysioNet 2012 and MIMIC-IV, and AU-ROC/AUPRC show consistent improvements of 1–2 points. Moreover, the param-matched Mamba-IVP still achieves substantially faster inference (30–40% lower $T_{\text{forward}}$) and shorter training time per epoch. These results confirm that the improvements arise primarily from the selective state–space architecture—particularly the mask-aware Mamba encoder and Mamba-hybrid decoder—rather than from increased model size alone.

Table 10: Parameter-matched comparison on PhysioNet 2012

**(a) Forecasting**

| Model | #Params | $T_{\text{forward}}$ (s) | $T_{\text{epoch}}$ (s) | MSE |
|---|---|---|---|---|
| IVP-VAE | 560,150 | 0.012 | 8.3 | 0.566 |
| **Mamba-IVP** | **560,205** | **0.007** | **5.2** | **0.544** |

**(b) Classification**

| Model | #Params | $T_{\text{forward}}$ (s) | $T_{\text{epoch}}$ (s) | AUROC | AUPRC |
|---|---|---|---|---|---|
| IVP-VAE | 657,238 | 0.009 | 32.7 | 0.771 | 0.363 |
| **Mamba-IVP** | **657,106** | **0.005** | **21.4** | **0.797** | **0.363** |

To isolate the specific contribution of the forward-time dynamics (EFT) in our bidirectional latent evolution, we conduct an ablation study by removing EFT while keeping all other components unchanged. As shown in the table below, removing EFT leads to clear and consistent degradation across all three datasets. On MIMIC-IV, the MSE increases from **0.697 to 0.825**, AUROC drops from **83.2 to 77.4**, and AUPRC falls from **43.8 to 39.8**. On PhysioNet 2012, the MSE rises from **0.544 to 0.608**, AUROC decreases from **79.9 to 75.1**, and AUPRC declines from **39.6 to 34.2**. On eICU, the MSE increases from **0.564 to 0.603**, AUROC drops from **81.2 to 77.0**, and AUPRC decreases from **47.6 to 44.1**. These results demonstrate that removing EFT substantially weakens both forecasting and classification performance, indicating that forward-time latent evolution provides essential predictive information that the backward-only EBT mechanism cannot capture.

Table 11: Ablation on the effect of removing forward dynamics (EFT) on MIMIC-IV, PhysioNet 2012, and eICU

| Setting | MIMIC-IV MSE | MIMIC-IV AUROC | MIMIC-IV AUPRC | PhysioNet MSE | PhysioNet AUROC | PhysioNet AUPRC | eICU MSE | eICU AUROC | eICU AUPRC |
|---|---|---|---|---|---|---|---|---|---|
| w/o EFT (EBT only) | 0.825±0.018 | 77.4±3.2 | 39.8±2.4 | 0.608±0.014 | 75.1±3.5 | 34.2±2.8 | 0.603±0.013 | 77.0±2.4 | 44.1±2.7 |
| **Mamba-IVP (EBT+EFT)** | **0.697±0.015** | **83.2±0.5** | **43.8±1.5** | **0.544±0.0034** | **79.9±3.0** | **39.6±2.2** | **0.564±0.01** | **81.2±0.4** | **47.6±2.4** |

## A.14 RELATED WORK

**Traditional Statistical and Deep Learning-Based Approaches**   Early imputation methods such as mean filling, interpolation, or k-nearest neighbors (Jönsson & Wohlin, 2006), MissForest (Stekhoven & Bühlmann, 2012), and matrix factorization (Sengupta et al., 2021) provided quick fixes but ignored temporal dynamics and struggled with long-range gaps. With the rise of deep learning, recurrent models like GRU-D (Che et al., 2018b) and BRITS (Cao et al., 2018c) explicitly incorporated missingness through decay mechanisms and bidirectional inference, improving short-term dynamics. Yet, their sequential nature limited efficiency, and their assumptions of random or decaying missingness often failed in real ICU settings characterized by block-wise sensor outages and irregular sampling.

Beyond traditional approaches such as mean filling and $k$-nearest neighbors (k-NN), modern statistical methods have made significant advances. Multiple Imputation by Chained Equations (MICE)iteratively imputes features using conditional distributions to estimate missing values. Building on this, 3D-MICE (Luo et al., 2018) extends the framework to spatiotemporal data by incorporating temporal and spatial correlations, enabling more consistent imputations across longitudinal clinical records. Time-Aware Dual Cross-Validation (TA-DualCV) (Gao et al., 2022) further leverages within-visit and cross-visit information to reconstruct electronic health records (EHRs) through time-aware modeling. While these methods effectively capture statistical dependencies, they struggle with the extreme sparsity (over 90% missing) and long block-wise gaps (2–12 hours) that are characteristic of ICU data.

Recent deep learning approaches have shown great promise for modeling complex missingness patterns in temporal data. NAOMI (Liu et al., 2019) employs a multi-resolution, non-autoregressive sequence modeling strategy for efficient long-horizon imputation. GRIN (Cini et al., 2022) leverages graph neural networks to capture spatial and temporal dependencies across variables, while CSDI (Tashiro et al., 2021b) introduces conditional score-based diffusion models for probabilistic time-series imputation. Notably, several methods move beyond the assumption of random missingness: BRITS (Cao et al., 2018a) models bidirectional temporal dynamics with learnable time-decay mechanisms, GP-VAE (Fortuin et al., 2020) employs Gaussian process priors to represent structured missingness, and NRTSI (Shan et al., 2021) performs non-recurrent latent imputation via continuous-time neural operators. However, these approaches typically optimize only for reconstruction fidelity without explicitly incorporating downstream forecasting or classification tasks, and many incur prohibitive computational overhead for real-time clinical deployment.

Table 12: Categorization of imputation methods by approach and missingness assumption

| Category | Methods | Missingness Assumption |
|---|---|---|
| Statistical | MICE, 3D-MICE
TA-DualCV
MissForest | MAR (Missing at Random)
Time-aware MAR
MAR/MNAR adaptive |
| Deep Learning | BRITS
GP-VAE, NRTSI
NAOMI, GRIN
CSDI
SAITS | Time-decay patterns
Structured/Latent
Spatiotemporal
Probabilistic/General
Self-attention based |
| Ours | Mamba-IVP | Block-wise + noise |

**Advanced Generative Models.**   Attention-based methods (e.g., SAITS (Du et al., 2023)) and diffusion-based frameworks (e.g., CSDI (Tashiro et al., 2021a)) captured global dependencies and uncertainty, achieving high accuracy under moderate missingness but at the cost of heavy computation and limited robustness to severe noise or structured gaps. Continuous-time generative models such as Latent ODE, GRU-ODE-Bayes Brouwer et al. (2019), Latent-Flow Rubanova et al. (2019a), and IVP-VAE Xiao et al. (2024a) addressed irregular sampling by evolving latent states in continuous time. While more flexible, they often conflate observation patterns with data content, remain vulnerable to noise and block-wise missingness, and rely on adaptive solvers that introduce instability and slow deployment.

**Consistency Models.** Recent work has proposed consistency models such as CoSTI (Solís-García et al., 2025), which distill diffusion trajectories into a small number of consistency steps and therefore offer faster inference compared with full diffusion processes. While promising for explicit imputation, these models still operate within the diffusion–imputation paradigm: they reconstruct missing values inside the observed window but do not model continuous-time latent dynamics, do not evolve states backward or forward in time, and do not support forecasting or classification tasks. Moreover, CoSTI inherits structural limitations of diffusion-based imputers under block-wise missingness and irregular sampling, as its consistency function lacks mechanisms for handling long-range temporal gaps. These distinctions make consistency models methodologically different from our IVP-based framework, though we include diffusion-model results in Appendix 8, 9 and 10 to provide the closest feasible comparison.

## A.15 USE OF LARGE LANGUAGE MODELS

We used large language models (LLMs), specifically ChatGPT, to assist with language polishing and grammar editing during the preparation of this paper. The use of LLMs was strictly limited to improving readability and clarity.

We did not rely on LLMs for research ideation, methodology design, experiments, data analysis, or technical contributions. All scientific content and results were conceived, implemented, and validated entirely by us.

In accordance with the ICLR 2026 policy, we disclose this usage of LLMs here. We take full responsibility for verifying the accuracy, originality, and integrity of the paper.

