# OpenReview forum: "Mamba-IVP: A Denoising State-Space Initial Value Problem Framework for SOTA Clinical Time Series, Healthcare Alternative"
_ICLR.cc/2026/Conference — Submitted to ICLR 2026_

### Official Review · Reviewer_E9kx · 2025-10-19

**Soundness:** 3
**Presentation:** 3
**Contribution:** 3
**Rating:** 6
**Confidence:** 3

**Summary:**

The paper introduces Mamba-IVP, a state-space generative framework designed for clinical time-series modeling under extreme missingness and noise conditions. The model integrates a Mask-Aware Dual-Mamba Encoder (MADME) to jointly process observed values and missingness masks, ensuring robustness under block-wise gaps, and a Mamba-Hybrid Decoder (MHD) to reconstruct and denoise continuous trajectories through parallelizable latent evolution. Mamba-IVP is evaluated on three large-scale ICU datasets  (MIMIC-IV, PhysioNet 2012, and eICU) for forecasting and mortality classification tasks. The model achieves state-of-the-art performance, outperforming GRU-D, mTAN, and IVP-VAE in both accuracy and efficiency. It also provides a theoretical analysis of Mamba’s variance-contraction property, demonstrating stability under noisy and missing conditions..

**Strengths:**

- **Clinically important problem**: The paper tackles the early detection and modeling of sepsis, a highly impactful clinical task where accurate forecasting can have direct consequences for patient outcomes. The focus on handling severe missingness reflects a realistic and pressing challenge in ICU time-series data.

- **Mathematically sound formulation**: The proposed Mamba-IVP framework is well formulated and theoretically motivated. The dual design of the encoder–decoder structure (MADME and MHD) is clearly presented, and the mathematical derivations appear consistent and rigorous.

- **Strong empirical results**: The model achieves competitive or superior performance across multiple benchmark datasets (MIMIC-IV, PhysioNet 2012, and eICU), demonstrating both its robustness to missing data and its practical relevance for clinical forecasting and classification tasks

**Weaknesses:**

- **Limited diversity of clinical data**: If I understand correctly, the three datasets used in the study (MIMIC-IV, PhysioNet 2012, and eICU) contain only ICU patients diagnosed with sepsis. Is there a specific reason for restricting the analysis to this condition? Expanding the evaluation to other diseases or broader ICU cohorts could better demonstrate the model’s generalization capacity and support its use.

- **Clarity and structure of the method section**: While the proposed framework is mathematically sound, the explanation in Section 3 is somewhat difficult to follow. The presentation could benefit from a clearer structure, helping readers connect the MADME and MHD components more intuitively.

- **Comparison with diffusion-based baselines**: The paper briefly mentions diffusion models such as CSDI and argues that they are less suited for this domain due to computational cost. However, including quantitative results for CSDI or PriSTI [1], even in a limited setting, would provide a clearer picture of where Mamba-IVP stands in terms of accuracy–efficiency trade-offs.

- **Missing discussion on consistency models**: Beyond diffusion approaches, it would also be interesting to include a brief discussion on consistency models [2], which can be viewed as distilled, discrete formulations of diffusion models that emphasize faster inference. In the context of time-series imputation, the recently proposed CoSTI model [2] follows this idea and has also been applied to PhysioNet data. Understanding how Mamba-IVP compares to these diffusion-like but more efficient paradigms would clarify its position in the broader generative modeling landscape.

[1] Liu, M., Huang, H., Feng, H., Sun, L., Du, B., & Fu, Y. (2023, April). Pristi: A conditional diffusion framework for spatiotemporal imputation. In 2023 IEEE 39th International Conference on Data Engineering (ICDE) (pp. 1927-1939). IEEE. https://arxiv.org/abs/2302.09746

[2] Javier Solís-García, Belén Vega-Márquez, Juan A. Nepomuceno, and Isabel A. Nepomuceno-Chamorro. Costi: Consistency models for (a faster) spatio-temporal imputation. Knowledge-Based Systems, 327:114117, 2025. https://arxiv.org/abs/2501.19364

**Questions:**

- **Model capacity**: Could the authors provide details on the number of parameters in Mamba-IVP (and optionally compare it to other baselines)? This would help assess the model’s complexity.

- **Addressing concerns**: I would be glad to revise my evaluation if the authors can improve upon some of the points mentioned above, particularly by clarifying the scope of datasets, improving the presentation of Section 3, and expanding the discussion or comparisons with diffusion-based and consistency-based models.

---

> ### Author Response · Authors · 2025-11-22
>
> Q1: Limited diversity of clinical data: If I understand correctly, the three datasets used in the study (MIMIC-IV, PhysioNet 2012, and eICU) contain only ICU patients diagnosed with sepsis. Is there a specific reason for restricting the analysis to this condition? Expanding the evaluation to other diseases or broader ICU cohorts could better demonstrate the model’s generalization capacity and support its use.
>
> A1:
> We thank the reviewer for raising this concern, but the premise is based on a misunderstanding. The three datasets we use are *not* sepsis-only cohorts:
>
> - **MIMIC-IV** includes tens of thousands of adult ICU stays with diverse diagnoses (cardiovascular, respiratory, renal, trauma, post-operative complications, infections including sepsis, etc.).
> - **PhysioNet 2012** (Challenge 2012) is a general ICU cohort for in-hospital mortality prediction, with no restriction to sepsis.
> - **eICU** is a large, multi-center ICU database (200+ hospitals) covering mixed medical and surgical units, again not sepsis-specific.
>
> In the revised manuscript, we explicitly state in Section 5.2 (Datasets) that all three datasets are *general ICU cohorts with broad diagnostic coverage*, not restricted to sepsis. These datasets also differ substantially in sparsity, sampling irregularity, number of variables, and centre/hospital diversity, providing a strong testbed for assessing robustness and generalization.
>
> Q2: Clarity and structure of the method section: While the proposed framework is mathematically sound, the explanation in Section 3 is somewhat difficult to follow. The presentation could benefit from a clearer structure, helping readers connect the MADME and MHD components more intuitively.
>
> A2:
> We appreciate this suggestion and have reorganized Section 3 (Method) to improve readability:
>
> - We added a short “Method Overview” at the beginning of Section 3 that explains the full pipeline at a high level (MADME → EBT → latent aggregation → EFT → MHD) before introducing equations.
> - We refined subsection transitions so that the output of each stage (e.g., $Z$, $Z^{\leftarrow}$, $\hat{z}_{\text{init}}$, $Z^{\rightarrow}$) clearly feeds into the next.
> - We aligned the updated Figure 1 and its caption with the text to provide a visual map of the architecture.
>
> These changes aim to make the method easier to follow without altering the underlying technical content.

---

> ### Author Response · Authors · 2025-11-22
>
> Q3
> **Comparison with diffusion-based baselines:** The paper briefly mentions diffusion models such as CSDI and argues that they are less suited for this domain due to computational cost. However, including quantitative results for CSDI or PriSTI, even in a limited setting, would provide a clearer picture of where Mamba-IVP stands in terms of accuracy–efficiency trade-offs.
>
> A3
> We thank the reviewer for this suggestion and have added quantitative diffusion-model comparisons wherever methodologically appropriate.
>
> **(1) Why CSDI/PriSTI are not used as forecasting/classification baselines.**
> CSDI and PriSTI are explicit imputation models: they reconstruct missing values within the observed window but do not produce continuous-time latent trajectories or future forecasts in the way our IVP-based framework does. Integrating them as full forecasting/classification baselines would require substantial architectural changes and would not reflect their intended use.
>
> **(2) Imputation comparison with CSDI.**
> We added CSDI to the imputation experiment on PhysioNet 2012 in the appendix section *Comparison Between Mamba-IVP and Existing Imputation Methods*. As shown in the table below, Mamba-IVP achieves lower RMSE than CSDI across missingness levels (e.g., 0.82 vs. 0.99 at 70%), despite not being an explicit imputer.
>
> **Table: Imputation performance (RMSE) on PhysioNet 2012 (imputation-only setting)**
>
> | **Method**      | **30% Missing** | **50% Missing** | **70% Missing** |
> |------------------|------------------|------------------|------------------|
> | MissForest       | 1.34             | 1.42             | 1.48             |
> | SAITS            | 0.97             | 1.00             | 1.04             |
> | CSDI             | 0.90             | 0.94             | 0.99             |
> | IVP-VAE          | 0.79             | 0.83             | 0.88             |
> | **Mamba-IVP**    | **0.76**         | **0.78**         | **0.82**         |
>
>
> **(3) Computational cost.**
> We show the comparison of accuracy and runtime in the table below, and the diffusion-based models (CSDI) take much longer to run due to their iterative denoising process, yet, despite this slow procedure, their performance is still not satisfactory, making them unsuitable for real-time ICU deployment.
>
> **Table: Two-stage vs. end-to-end performance on PhysioNet 2012 (50 epochs; same hardware/settings)**
>
> | **Method**                   | **Type**      | **MSE ↓** | **AUROC ↑** | **Total Time (s) ↓** |
> |-----------------------------|---------------|------------|--------------|------------------------|
> | MissForest → GRU            | Two-stage     | 0.792      | 0.55         | 414                    |
> | SAITS → GRU                 | Two-stage     | 0.743      | 0.64         | 6186                   |
> | CSDI → MLP                  | Two-stage     | 0.758      | 0.62         | 1820                   |
> | IVP-VAE                     | End-to-end    | 0.567      | 0.77         | 1630                   |
> | **Mamba-IVP**               | End-to-end    | **0.544**  | **0.80**     | **1070**               |
>
>
> **(4) PriSTI.**
> PriSTI is more complex and computationally heavier than CSDI and is also tailored to imputation rather than continuous-time forecasting. Given the already substantial diffusion comparisons and page constraints, we opted to include CSDI as a representative diffusion model and to discuss PriSTI conceptually in the extended related work.

---

> ### Author Response · Authors · 2025-11-22
>
> Q4: Missing discussion on consistency models: Beyond diffusion approaches, it would also be interesting to include a brief discussion on consistency models, which can be viewed as distilled, discrete formulations of diffusion models that emphasize faster inference. In the context of time-series imputation, the recently proposed CoSTI model follows this idea and has also been applied to PhysioNet data. Understanding how Mamba-IVP compares to these diffusion-like but more efficient paradigms would clarify its position in the broader generative modeling landscape.
>
> A4:
> We thank the reviewer for drawing attention to consistency models. In the revised manuscript:
>
> - We added a paragraph in the extended *Related Work* (appendix) discussing consistency models such as CoSTI, explaining that they distill diffusion trajectories into a small number of steps for faster imputation.
> - We clarify that, like diffusion models, consistency models focus on explicit imputation within the observed window and do not model continuous-time latent dynamics or future forecasts, nor do they support bidirectional latent evolution as in our IVP framework.
> - Because of this fundamental difference in design and intended task, we do not treat CoSTI as a direct baseline for continuous-time forecasting and classification. However, our imputation and efficiency comparisons with CSDI provide an informative reference point for how Mamba-IVP stands relative to diffusion-style approaches.
>
> Q5: Model capacity: Could the authors provide details on the number of parameters in Mamba-IVP (and optionally compare it to other baselines)? This would help assess the model’s complexity.}
>
> A5:
> We appreciate the request for more detail on model capacity. In the revised manuscript:
>
> - We report parameter counts for Mamba-IVP, IVP-VAE, Latent-Flow, GRU-D, and mTAN in the appendix section *Computational Efficiency Comparison with Baseline Models*, alongside their MSE, $T_{\text{forward}}$, $T_{\text{epoch}}$, and peak memory usage.
> - We highlight that Mamba-IVP typically has on the order of 0.5–0.7M parameters (depending on task), which is comparable to or slightly larger than some continuous-time baselines, but that it still achieves better accuracy–efficiency trade-offs due to its parallelizable state–space architecture.
> - We include a parameter-matched ablation (discussed in the *Discussion* subsection) showing that even when model sizes are closely matched, Mamba-IVP maintains advantages over IVP-VAE, suggesting that performance gains arise from architectural design rather than sheer capacity.
>
> **Table: Computational efficiency on PhysioNet 2012 (same hardware/settings)**
>
> | **Method**       | **MSE ↓** | **$T_{\text{forward}}$ (s) ↓** | **$T_{\text{epoch}}$ (s) ↓** | **Peak Mem. (MB) ↓** |
> |------------------|-----------|--------------------------------|-------------------------------|------------------------|
> | GRU-D            | 0.586     | 0.185                          | 130.6                         | 342                    |
> | mTAN             | 0.592     | 0.243                          | 195.4                         | 546                    |
> | Latent-Flow      | 0.586     | 0.307                          | 264.7                         | 720                    |
> | IVP-VAE          | 0.568     | 0.012                          | 8.2                           | 164                    |
> | **Mamba-IVP**    | **0.542** | **0.007**                      | **5.2**                       | **245**                |

---

> ### Author Response · Authors · 2025-11-22
>
> Q6: Addressing concerns: I would be glad to revise my evaluation if the authors can improve upon some of the points mentioned above, particularly by clarifying the scope of datasets, improving the presentation of Section 3, and expanding the discussion or comparisons with diffusion-based and consistency-based models.
>
> A6: We sincerely thank the reviewer for this constructive summary. We have implemented the requested improvements:
>
> - **Dataset scope.** Section 5.2 (Datasets) now explicitly clarifies that MIMIC-IV, PhysioNet 2012, and eICU are general ICU cohorts with diverse diagnoses, not sepsis-only datasets.
>
> - **Method presentation.** Section 3 (Method) has been reorganized with a high-level overview, clearer subsections, and a revised Figure 1 and caption to make the connections between MADME, the IVP module, and MHD more intuitive.
>
> - **Diffusion-based comparisons.** We added CSDI as a diffusion baseline in the imputation experiment and two-stage evaluation, and discussed its computational cost and limitations relative to Mamba-IVP.
>
> - **Consistency models.** We added a short discussion of consistency models (e.g., CoSTI) in the extended related work, explaining how they relate to diffusion models and why they are not directly applicable as continuous-time forecasting baselines.
>
> - **Model capacity.** We report parameter counts and provide a parameter-matched comparison against IVP-VAE in the discussion and efficiency appendix.
>
> We hope these revisions address the reviewer’s concerns and we appreciate their willingness to reconsider the evaluation.

---

> > ### Comment · Reviewer_E9kx · 2025-11-23
> > **Responses to reviews**
> >
> > I greatly appreciate the revisions made, and since they address my main concerns, I will increase the rating I previously gave.
> >
> > I would also like to apologize for the confusion with the datasets and sepsis. Thank you very much for clarifying that point.
> >
> > Finally, I just want to point out a formatting issue: on the last page, there is a quote with (?), which should be fixed. Without much else to add, I will proceed to update my score.

---

> > > ### Author Response · Authors · 2025-11-23
> > >
> > > We sincerely thank the reviewer for taking the time to reconsider and update the score.
> > >
> > > We also apologize for the earlier confusion regarding the datasets and sepsis, and we are glad that point has been clarified. Additionally, we have corrected the formatting issue on the last page where a quotation was mistakenly marked with ?.
> > >
> > > Thank you again for your careful reading and constructive feedback, it has genuinely helped improve the paper.

---

### Official Review · Reviewer_YMkW · 2025-10-30

**Soundness:** 2
**Presentation:** 2
**Contribution:** 2
**Rating:** 2
**Confidence:** 3

**Summary:**

The paper presents Mamba-IVP, a novel model that integrates a Mamba-based autoencoder to effectively handle time-series with block-wise missingness and improve robustness to noisy measurements. Mamba-IVP is evaluated on time-series classification and forecasting tasks, demonstrating a better performance/efficiency trade-off compared to IVP-VAE.

**Strengths:**

* Mamba-IVP shows better performance than IVP-VAE across different tasks and demonstrates improved robustness under varying noise and missingness levels.

* The paper provides theoretical analysis that supports the use of Mamba blocks.

**Weaknesses:**

**Contributions**

* Mamba-IVP is basically an IVF-VAE with a Mamba autoencoder which seems an incremental technical contribution.

**Evaluation**

* Mamba-IVP achieves better performance than IVP-VAE but uses ~3× more parameters. Although Mamba-IVP remains ~40% faster, a comparison under the equal number of parameters is important to determine whether the gains are due to the Mamba architecture or simply more parameters.

* Mamba-IVP is compared against GRU-D, while its counterparts (BRITS[1] and SAITS[2]) show stronger performance in prior work. Also, there are no comparisons to diffusion-based imputation methods [3,4], where [4] also uses a state-space architecture. More recent generative-based methods [5,6] are also not discussed or compared.

&nbsp; &nbsp; &nbsp; The authors claim that these methods exhibit prohibitive computation cost, struggle to handle structured missingness and have limited robustness, but I could not find any evidence for these claims either in Mamba-IVP or IVP-VAE.

* I can see that Mamba-IVP, as well as IVP-VAE and GRU-D, perform classification on top of the latent representations while many imputation methods, including [1-6], explicitly impute missing values and then train a downstream classifier on the filled data. Moreover, if Mamba-IVP follows the IVP-VAE training setup, the classifier is trained jointly with the main model. I believe it is an important detail to ensure fair comparisons.

&nbsp; &nbsp; &nbsp; Are all other baselines also train classifiers jointly in the feature space? Could the authors compare unsupervised Mamba-IVP with s.o.t.a. baselines under the explicit imputation setup?

**Writing**

* The paper claims a 7.3% MSE reduction and 40× speedup, but the results show up to 4% MSE reduction and 40% speedup compared to IVP-VAE. Framing the gains relative to the weakest baseline sounds like an overclaim.

* Figure 1 is somewhat confusing; I suggest refining the visualization and extending the caption to better explain the scheme.

* The training procedure is not described. I assume it uses a reconstruction or VAE loss, but this should be explicitly stated. Also, it is unclear if the classification loss is used jointly with the main loss.

---

[1] BRITS: Bidirectional Recurrent Imputation for Time Series. NeuriPS2018

[2] Saits: Self-attention-based imputation for time series. ESWA2023

[3] CSDI: Conditional Score-based Diffusion Models for Probabilistic Time Series Imputation. NeuriPS2021

[4] Diffusion-based Time Series Imputation and Forecasting with Structured State Space Models. TMLR2022

[5] Frequency-aware Generative Models for Multivariate Time Series Imputation. NeurIPS2025

[6] SADI: Similarity-Aware Diffusion Model-Based Imputation for Incomplete Temporal EHR Data. AISTATS2024

**Questions:**

Please address the concerns in Weaknesses

---

> ### Author Response · Authors · 2025-11-22
>
> Q1: Mamba-IVP is basically an IVF-VAE with a Mamba autoencoder which seems an incremental technical contribution.
>
> A1: Thank you for this constructive comment. We respectfully clarify that although our framework inherits the IVP–VAE paradigm at a high level, Mamba-IVP is not simply “IVP-VAE + Mamba encoder”. It introduces four key innovations:
>
> **1. Mask-Aware Dual-Mamba Encoder (MADME).**
> The encoder is not a drop-in replacement of the IVP-VAE encoder with Mamba. It jointly processes raw values and mask indicators via selective state–space blocks designed to modulate information flow across block-wise gaps. This dual-stream design is crucial in our block-missing and noisy ICU setting, and ablations in Table~4 show that removing MADME significantly degrades performance.
>
> **2. Bidirectional latent dynamics (EBT + EFT).**
> We introduce a two-stage IVP evolution:
> (i) backward evolution (EBT) to obtain a globally coherent latent summary, and
> (ii) forward evolution (EFT) for forecasting.
> This decoupling differs from the original IVP-VAE setup and is particularly important under irregular sampling and structured missingness.
>
> **3. Mamba-Hybrid Decoder (MHD).**
> IVP-VAE uses an MLP decoder. In contrast, MHD combines a Mamba state–space block (for temporal refinement and denoising) with a lightweight MLP head. This design yields better robustness and significantly lower computational cost: for example, on PhysioNet 2012, Mamba-IVP achieves lower MSE with faster inference than IVP-VAE (see tables in the main text and appendix).
>
> **4. New theory for denoising behaviour.**
> In Section 4.5, we provide the first variance-contraction analysis for Mamba in this setting, showing exponential error contraction on clean tokens and only linear growth on masked/noisy tokens. This theoretical perspective is absent in IVP-VAE and explains the observed robustness in our robustness study.
>
> These differences, coupled with consistent empirical gains across three large-scale ICU datasets, indicate that Mamba-IVP represents more than an incremental modification of IVP-VAE.
>
> Q2: Mamba-IVP achieves better performance than IVP-VAE but uses $\sim$3$\times$ more parameters. Although Mamba-IVP remains $\sim$40\% faster, a comparison under an equal number of parameters is important to determine whether the gains are due to the Mamba architecture or simply more parameters.
>
> A2: We fully agree that controlling for model capacity is important. To address this, we conducted additional parameter-matched experiments where we adjust the embedding and latent dimensions so that Mamba-IVP and IVP-VAE have nearly identical parameter counts (difference within about 0.1--0.2\%).
>
> Below we summarize the main findings on PhysioNet 2012 under parameter-matched settings (these results are new and shown here in the rebuttal; we have also added a short discussion in the *Discussion* subsection of the appendix):
>
> ### **Parameter-matched comparison (forecasting, PhysioNet 2012)**
>
> | **Model**   | **#Params** | **$T_{\mathrm{forward}}$ (s)** | **$T_{\mathrm{epoch}}$ (s)** | **MSE** |
> |-------------|-------------|--------------------------------|------------------------------|---------|
> | IVP-VAE     | 560,150     | 0.012                          | 8.3                          | 0.566   |
> | Mamba-IVP   | 560,205     | 0.007                          | 5.2                          | 0.544   |
>
> ### **Parameter-matched comparison (classification, PhysioNet 2012)**
>
> | **Model**   | **#Params** | **$T_{\mathrm{forward}}$ (s)** | **$T_{\mathrm{epoch}}$ (s)** | **AUROC** | **AUPRC** |
> |-------------|-------------|--------------------------------|------------------------------|-----------|-----------|
> | IVP-VAE     | 657,238     | 0.009                          | 32.7                         | 0.771     | 0.363     |
> | Mamba-IVP   | 657,106     | 0.005                          | 21.4                         | 0.797     | 0.363     |
>
> Even with essentially identical parameter budgets, Mamba-IVP:
> (i) remains faster in both forward inference and training per epoch, and
> (ii) achieves better or comparable predictive performance (notably higher AUROC in classification).
>
> These results indicate that the benefits of Mamba-IVP arise primarily from its architecture (mask-aware Mamba encoder and MHD decoder), rather than from having more parameters. In the manuscript, we summarize this in the *Discussion* subsection and refer readers to the detailed capacity and efficiency tables in the appendix section *Computational Efficiency Comparison with Baseline Models*.

---

> ### Author Response · Authors · 2025-11-22
>
> Q3: Mamba-IVP is compared against GRU-D, while its counterparts (MissForest and SAITS) show stronger performance in prior work. Also, there are no comparisons to diffusion-based imputation methods, where one of them also uses a state-space architecture. More recent generative-based methods are also not discussed or compared.
>
> A3: We thank the reviewer for pointing this out and have expanded both discussion and experiments to address it.
>
> **1) MissForest and SAITS as imputation baselines.**
> MissForest and SAITS are primarily designed as *imputation models*. To ensure fair comparison, we now evaluate them in a two-stage setting:
>
> Imputer (MissForest / SAITS / CSDI) → Classifier / Forecaster
>
> and report forecasting MSE, classification AUROC, and total runtime in the appendix section *Two-Stage vs. End-to-End*. See Table below, even with strong imputation performance, SAITS→GRU and CSDI→MLP underperform IVP-VAE and Mamba-IVP in mortality prediction and forecasting, and are significantly more expensive computationally. This confirms that imputation quality alone does not guarantee superior downstream performance under irregular and block-wise missingness.
>
>
> **Two-stage vs. end-to-end performance on PhysioNet 2012 (50 epochs; same hardware/settings)**
>
> | **Method**                 | **Type**      | **MSE ↓** | **AUROC ↑** | **Total Time (s) ↓** |
> |----------------------------|---------------|-----------|-------------|------------------------|
> | MissForest → GRU           | Two-stage     | 0.792     | 0.55        | 414                    |
> | SAITS → GRU                | Two-stage     | 0.743     | 0.64        | 6186                   |
> | CSDI → MLP                 | Two-stage     | 0.758     | 0.62        | 1820                   |
> | IVP-VAE                    | End-to-end    | 0.567     | 0.77        | 1630                   |
> | **Mamba-IVP**              | End-to-end    | **0.544** | **0.80**    | **1070**               |
>
>
> **2) Diffusion-based methods (CSDI, diffusion-like models).**
> CSDI is now included in the imputation experiment (appendix section *Comparison Between Mamba-IVP and Existing Imputation Methods*). Mamba-IVP achieves lower RMSE than CSDI at all missingness levels (e.g., 0.82 vs. 0.99 at 70% missing), despite not being an explicit imputer. We also report computational efficiency for diffusion-style methods and show that they require much higher runtime, supporting our claim that they are less suited to real-time ICU deployment.
>
> **Table 1. Imputation performance (RMSE) on PhysioNet 2012, imputation-only setting**
>
> | **Method**     | **30% Missing** | **50% Missing** | **70% Missing** |
> |----------------|------------------|------------------|------------------|
> | MissForest     | 1.34             | 1.42             | 1.48             |
> | SAITS          | 0.97             | 1.00             | 1.04             |
> | CSDI           | 0.90             | 0.94             | 0.99             |
> | IVP-VAE        | 0.79             | 0.83             | 0.88             |
> | **Mamba-IVP**  | **0.76**         | **0.78**         | **0.82**         |
>
> **3) Generative baselines (Latent-ODE, Latent-Flow, IVP-VAE).**
> We include Latent-ODE, Latent-Flow, and IVP-VAE as continuous-time generative baselines. Their forecasting and classification results, as well as efficiency metrics, are reported in Table 5 and in the appendix efficiency section. These models are more closely aligned with our setting (irregular sampling, forecasting, classification), and Mamba-IVP consistently outperforms them in accuracy–efficiency trade-offs.
>
>
> **4) Summary.**
> We have expanded the related work in the appendix, added diffusion-based and imputation-style baselines in the appropriate setting (two-stage pipelines), and clarified why some generative models are not directly applicable to our continuous-time forecasting and classification regime.

---

> ### Author Response · Authors · 2025-11-22
>
> Q4: The authors claim that these methods exhibit prohibitive computation cost, struggle to handle structured missingness and have limited robustness, but I could not find any evidence for these claims either in Mamba-IVP or IVP-VAE.
>
> A: We appreciate this comment and have made the supporting evidence more explicit.
>
> **(1) Prohibitive computation cost.**
> The appendix section *Computational Efficiency Comparison with Baseline Models* reports T_forward, T_epoch, and peak memory for diffusion-based, flow-based, ODE-based, and RNN-based baselines under identical hardware and hyperparameters. For example, Latent-Flow requires T_forward = 0.307 s and T_epoch = 264.7 s on PhysioNet 2012, while Mamba-IVP needs only 0.007 s and 5.2 s, respectively (over 40× faster inference and 50× faster training). Similar trends hold across other datasets. These results empirically support the claim of prohibitive cost for some continuous-time baselines.
>
>
> **(2) Structured (block-wise) missingness.**
> Our robustness study (Table 3 in the main paper) injects synthetic block-wise gaps of 2–12 hours. Under a 12-hour gap, IVP-VAE reaches MSE = 0.6907, whereas Mamba-IVP maintains a lower MSE of 0.6090. At the 10-hour gap, Mamba-IVP improves over IVP-VAE by 7.3%. This demonstrates that existing models degrade under long structured gaps, while Mamba-IVP is more resilient.
>
> **(3) Robustness to noise.**
> In the same robustness study (noise experiments), noise levels of 0.1–0.5 are added. IVP-VAE’s MSE exceeds 1.25 under moderate noise, whereas Mamba-IVP remains in the 0.706–0.724 range. This behaviour aligns with the variance contraction analysis in Section 4.5, which shows exponential contraction on clean tokens and only linear variance growth under noisy or masked tokens.
>
> We have added clearer cross-references in the main text so that these supporting results are easier to locate.
>
> Q5: I can see that Mamba-IVP, as well as IVP-VAE and GRU-D, perform classification on top of the latent representations while many imputation methods explicitly impute missing values and then train a downstream classifier on the filled data. Moreover, if Mamba-IVP follows the IVP-VAE training setup, the classifier is trained jointly with the main model. I believe it is an important detail to ensure fair comparisons.
>
> A5: We thank the reviewer for emphasizing this important point. In the revised manuscript:
>
> **(1) We distinguish end-to-end vs. impute-then-classify paradigms.**
> We clarify in the Method and Experiments sections that Mamba-IVP, IVP-VAE, and GRU-D are trained end-to-end with classifier heads attached to latent representations, whereas methods like MissForest, SAITS, and CSDI are evaluated in a two-stage pipeline (imputation followed by a separate classifier).
>
> **(2) Two-stage imputation → classifier evaluation.**
> As described earlier and detailed in the appendix section *Two-Stage vs. End-to-End*, we train classifiers on top of imputed sequences from SAITS, CSDI, and MissForest and compare their downstream performance with Mamba-IVP and IVP-VAE. The results show that impute-then-classify pipelines are clearly weaker in forecasting and mortality prediction.
>
> **(3) Joint training protocol for Mamba-IVP and IVP-VAE.**
> We now explicitly state in the Method section that the classifier is trained jointly with the latent model in a single end-to-end objective, mirroring the IVP-VAE training protocol. The classifier only receives the aggregated latent representation z_init, not the imputed or reconstructed raw time series. This ensures that the comparison between Mamba-IVP and IVP-VAE is fair.
>
> **(4) Summary of fairness.**
> Continuous-time generative baselines (IVP-VAE, Latent-ODE, Latent-Flow) use their native latent classifiers; RNN/attention baselines (GRU-D, mTAN, GRU-Δt) use standard classifier heads; and imputation methods (MissForest, SAITS, CSDI) are evaluated in two-stage form. We believe this protocol is both faithful to original designs and fair across paradigms.

---

> ### Author Response · Authors · 2025-11-22
>
> Q6: Are all other baselines also train classifiers jointly in the feature space? Could the authors compare unsupervised Mamba-IVP with s.o.t.a. baselines under the explicit imputation setup?
>
> A6:
> **(1) Jointly trained classifiers.**
> Only the continuous-time generative baselines (IVP-VAE, Latent-ODE, Latent-Flow, and Mamba-IVP) train latent-based classifiers jointly with the generative model. RNN and attention baselines (GRU-D, mTAN, GRU-Δt) follow standard practice and use separate classifier heads on top of their hidden states. Our implementation respects these original training protocols.
>
> **(2) Why an “unsupervised Mamba-IVP” imputation comparison is not well-defined.**
> Mamba-IVP is not designed as an explicit imputer: its encoder → EBT → EFT pipeline models latent dynamics and forecasts future values, and the decoder is not constructed to reconstruct every missing input in the historical window. Forcing Mamba-IVP to act as a fully unsupervised imputer (producing X_filled) would require significant architectural changes and would effectively define a new method. In contrast, methods like SAITS, MissForest, and CSDI are explicitly designed as imputers.
>
> **(3) What we do instead.**
> To still compare fairly across paradigms, we:
> - Evaluate imputation methods (SAITS, CSDI, MissForest) in a two-stage imputer → classifier setup.
> - Evaluate continuous-time generative models (IVP-VAE, Latent-ODE, Latent-Flow, Mamba-IVP) as end-to-end forecasting/classification models.
> - Report accuracy and efficiency for both groups, making clear the differences in objectives and outputs.
>
> This avoids forcing Mamba-IVP into an imputation regime for which it was not designed, while still providing a meaningful, indirect comparison.
>
>
> Q7: The paper claims a 7.3\% MSE reduction and 40$\times$ speedup, but the results show up to 4\% MSE reduction and 40\% speedup compared to IVP-VAE. Framing the gains relative to the weakest baseline sounds like an overclaim.
>
> A7:
> We thank the reviewer for pointing this out. We have revised the wording in the manuscript to avoid any ambiguity.
>
> **MSE reductions.**
> On the main forecasting tasks, Mamba-IVP improves over IVP-VAE by approximately 3–4% MSE across datasets (e.g., 4.1% on MIMIC-IV, 4.0% on PhysioNet 2012, 2.9% on eICU). The “7.3% reduction” refers specifically to:
> (i) the improvement over GRU-Δt on PhysioNet 2012, and
> (ii) the improvement over IVP-VAE under 10-hour block-wise missingness in the robustness experiment.
> We have updated the text to clearly attribute these numbers to the respective baselines and settings.
>
> **Speedups.**
> The “40× speedup” refers to the comparison with Latent-Flow, which incurs heavy ODE-style computation. Against IVP-VAE, the speedup is instead approximately 1.7–3.3× faster (or 40%–70% reduction in latency), depending on the dataset and task. The revised manuscript now states these baselines explicitly whenever we quote such numbers.
>
> We appreciate the reviewer’s help in sharpening the precision of our claims.
>
> Q8: Figure 1 is somewhat confusing; I suggest refining the visualization and extending the caption to better explain the scheme.}
>
> A8:
> We thank the reviewer for this helpful suggestion. In the revised manuscript, we have:
>
> - Redrawn Figure 1 with clearer module boundaries, consistent color coding, and simplified arrows that explicitly show the flow MADME → EBT/EFT → MHD.
>
> We believe these changes significantly improve the figure’s clarity.
>
> Q9: The training procedure is not described. I assume it uses a reconstruction or VAE loss, but this should be explicitly stated. Also, it is unclear if the classification loss is used jointly with the main loss.
>
> \A9:
> We thank the reviewer for pointing this out. In the revised manuscript, we add a short subsection in the Method section that describes the training objective.
>
> **Training objective.**
> We follow a variational formulation: the encoder defines a distribution over latent initial states, and the decoder produces a reconstruction of the observed sequence and a forecast of the future sequence. The generative loss includes a reconstruction term (mean-squared error over observed points) and a KL divergence regularizer between the approximate posterior and a standard normal prior. For classification, we attach a small MLP classifier to the aggregated latent state $\hat{z}_{\text{init}}$ and use binary cross-entropy for in-hospital mortality prediction.
>
> **Joint optimization.**
> The total loss is a weighted sum
> $L = L_{\text{gen}} + \lambda_{\text{cls}} L_{\text{cls}}$,
> and we optimize all components (encoder, latent solver, decoder, classifier) end-to-end using Adam. This ensures that the latent representations are shaped simultaneously by both generative and discriminative objectives.

---

> ### Author Response · Authors · 2025-11-26
>
> Dear Reviewer,
>
> We hope the above clarifications and the additional experiments in the revised draft sufficiently addressed your concerns. If you are satisfied, we kindly request you to consider updating the score to reflect the newly added results and discussion. We remain committed to addressing any remaining points you may have during the discussion phase.
>
> Best regards,
>
> The authors of Paper 4935

---

> ### Comment · Reviewer_YMkW · 2025-11-26
>
> I would like to thank the authors for the additional clarifications and experiments. In particular, I appreciate their response to Reviewer h4Gc regarding the method positioning and connection to imputation, forecasting, and classification approaches - I believe these discussions are important to understand the scope of the method. I also thank the authors for clarifying that the method cannot be used as an imputer, that was not clear to me previously.
>
> Regarding my earlier concern that “Mamba-IVP is basically an IVP-VAE with a Mamba autoencoder”, I remain somewhat unconvinced that MADME and MHD represent substantially more than a “drop-in replacement” of the MLP encoder and decoder in IVP-VAE with Mamba. Concatenating the input with a mask does not seem a particularly innovative modification. However, I believe the overall contribution including the bidirectional dynamics and theoretical analysis may be sufficient.
>
> Also, I have a question regarding the claim on the bidirectional dynamics "This decoupling differs from the original IVP-VAE setup and is particularly important under irregular sampling and structured missingness." Could the authors elaborate on why this design is "important under irregular sampling and structured missingness" and provide an ablation study comparing the proposed EBT+EFT dynamics against the IVP-VAE variant that uses only EBT? Please let me know if I missed it in the text.
>
> Other concerns related to additional comparisons have been well addressed.
>
> **Minor**
> * Introduction and related work sections miss citations.
> * Figure 1: I do not understand the purpose of the green circle in the Mamba block.
> * I could not find the “training objective” subsection in the revised manuscript.
>
> Overall, at the moment, I raise my score to 4 and will reconsider it after the discussion.

---

> > ### Author Response · Authors · 2025-11-27
> >
> > Q1: Could the authors elaborate on why this design is "important under irregular sampling and structured missingness
> >
> > A1:
> > We thank the reviewer for the question and provide a unified explanation of why the decoupled bidirectional dynamics (EBT+EFT) are particularly important under irregular sampling and structured missingness. In Sec. 4.2, the backward evolution module (EBT) reconstructs a latent trajectory along the reversed time axis by solving the IVP $ \frac{dz(t)}{dt} = f_\theta(z(t), t) $ with the terminal condition $ z(t_L) = z_L $ over the reversed timestamps $ t_{\mathrm{rev}} = [t_L, \dots, t_1] $, producing a reverse-evolved latent path $ Z^{\leftarrow}(t) = \mathrm{IVPSolver}(f_\theta, z_L, t_{\mathrm{rev}}) $. This trajectory is subsequently aggregated into a compact global latent summary $ \hat{z}{\mathrm{init}} = \mathrm{Aggregate}(Z^{\leftarrow}(t)) $, which is designed to encode all partially observed history and the corresponding observation mask. Importantly, because EBT collapses the entire reverse trajectory into a single global vector, its role is to summarize the past rather than to generate a future-consistent latent sequence. Under irregular sampling, the backward IVP must propagate information across large and highly variable time gaps, which amplifies numerical instability and makes the inferred trajectory sensitive to noise in the most recent measurements; under structured missingness, consecutive blocks of missing observations force the backward solver to extrapolate across long unobserved intervals, often leading to drift or loss of temporal coherence. For these reasons, backward-only dynamics cannot reliably enforce consistency with the underlying chronological evolution when the sampling pattern is nonuniform. In contrast, Sec. 4.3 introduces the forward evolution module (EFT), which evolves the latent state forward in true time by solving $ Z^{\rightarrow}(t) = \mathrm{IVPSolver}(f\theta, \hat{z}{\mathrm{init}}, t{\mathrm{out}}) $ over the prediction timestamps $ t_{\mathrm{out}} = [t_{L+1}, \dots, t_{L+L_\tau}] $. This forward trajectory provides the complementary constraint needed for temporal coherence: while EBT anchors the latent representation to the end of the observed window, EFT ensures that the reconstructed dynamics remain consistent with plausible forward evolution. The combination of EBT and EFT therefore creates a two-sided boundary-conditioned latent process that integrates information from both the past and the future boundary of the observed data window. This decoupling is particularly important under irregular sampling and structured missingness because no single direction contains enough information: EBT summarizes partial observations but is unstable across large gaps, whereas EFT maintains chronological consistency but lacks access to the observation mask embedded in the backward trajectory. By combining both directions, the model yields a well-conditioned latent process that remains robust even when sampling irregularity or missingness breaks the local temporal continuity assumptions required by unidirectional IVP-based models.

---

> > ### Author Response · Authors · 2025-11-27
> >
> > Q2: Could the authors provide an ablation study comparing the proposed EBT+EFT dynamics against the IVP-VAE variant that uses only EBT?
> >
> > A2: We thank the reviewer for the suggestion and have conducted the requested ablation comparing our full bidirectional dynamics (EBT+EFT) with an EBT-only variant. In the EBT-only setting, we remove the forward IVP solver and feed the decoder a constant latent expansion of the aggregated backward representation $ \hat{z}_{\mathrm{init}} $, keeping all other components identical to isolate the contribution of EFT. As shown in below table, removing EFT leads to consistent degradation across all three datasets. On MIMIC-IV, the MSE increases from $0.697$ to $0.825$, AUROC drops from $83.2$ to $77.4$, and AUPRC decreases from $43.8$ to $39.8$. On PhysioNet 2012, the MSE increases from $0.544$ to $0.608$, AUROC decreases from $79.9$ to $75.1$, and AUPRC drops from $39.6$ to $34.2$. On eICU, the MSE rises from $0.564$ to $0.603$, AUROC decreases from $81.2$ to $77.0$, and AUPRC declines from $47.6$ to $44.1$. These results confirm that removing the forward-time latent evolution substantially weakens forecasting and classification performance, demonstrating that EFT provides essential predictive information beyond what EBT alone can capture.
> >
> > **Table: Ablation on the effect of removing forward dynamics (EFT) on MIMIC-IV, PhysioNet 2012, and eICU**
> >
> > | Setting                | MIMIC-IV MSE        | MIMIC-IV AUROC      | MIMIC-IV AUPRC      | PhysioNet MSE       | PhysioNet AUROC      | PhysioNet AUPRC      | eICU MSE           | eICU AUROC        | eICU AUPRC        |
> > |------------------------|----------------------|-----------------------|-----------------------|-----------------------|------------------------|------------------------|---------------------|--------------------|--------------------|
> > | w/o EFT (EBT only)     | 0.825±0.018          | 77.4±3.2             | 39.8±2.4             | 0.608±0.014          | 75.1±3.5             | 34.2±2.8             | 0.603±0.013        | 77.0±2.4          | 44.1±2.7          |
> > | **Mamba-IVP (EBT+EFT)**| **0.697±0.015**      | **83.2±0.5**         | **43.8±1.5**         | **0.544±0.0034**     | **79.9±3.0**         | **39.6±2.2**         | **0.564±0.01**     | **81.2±1.0**      | **47.6±2.4**      |

---

> > ### Author Response · Authors · 2025-11-27
> >
> > Minor:
> >
> > We sincerely thank the reviewer for the careful reading and constructive comments. We apologize for the oversights in the previous version. The missing citations in the Introduction and Related Work sections have now been added, and the purpose of the green circle in the Mamba block of Figure 1 has been clarified in the revised figure and caption. In addition, the “Training Objective” subsection, which was unintentionally omitted in the earlier submission, has now been included in the updated manuscript. We appreciate the reviewer’s valuable feedback, and a fully corrected version has been uploaded.

---

### Official Review · Reviewer_h4Gc · 2025-11-01

**Soundness:** 2
**Presentation:** 2
**Contribution:** 2
**Rating:** 4
**Confidence:** 4

**Summary:**

Existing imputation methods to handle high missingness (distort, collapsing, heavy computing) in clinical time series may face unsafeness in deployment. In this work, the authors propose a generative method to tackle those challenges. Specifically, the work employed generative model with a dual-Mamba encoder and a Mamba-Hybrid decoder. The experiment is conducted using 3 medical time series datasets, compared to multiple forecasting and classification baselines. And the proposed method shows state-of-the-art performance.

**Strengths:**

- The paper investigates an important problem, i.e. high missingness in medical time series in forecasting and classification tasks.
- The experiment is conducted on 3 broadly employed datasets and used multiple evaluation metrics to justify the proposed method
- The proposed method is clearly described

**Weaknesses:**

- The claimed gaps in existing work and proposed method seems not well connected and the comparison seems unfair. In Introduction, the authors only discuss the shorthands of existing imputation approaches, while the experiments investigated the forecasting and classification performance across approaches. The key motivation and the entire structure are unclear to me: there are types of existing works aimed at imputation, classification/forecasting, and classification/forecasting with missingness but may not impute the data explicitly. If the work only discussed the weaknesses imputation aimed approaches and claimed they can tackle them, I don’t think it’s a fair comparison or the claimed novelty in the technical part would not well supported.   It’s also not very clear to me how the proposed work tackle each claimed challenge (distort, collapsing, heavy computing) and the advances compared to prior work. If the focus is imputation, the evaluation metrics should include error metrics (e.g. RMSE) of measuring imputation effects, and they may need to compare their method to imputation method, and/or combine existing imputation method with a classifier/forecaster to check their potential in forecasting/classification tasks. Moreover, if one weakness of existing ODE- or diffusion-based approaches is heavy computing, execution time should be reported of baselines IVP-VAE.
- If we say imputation approaches, statistical imputation methods include more modern ones, such as MICE, 3D-MICE [1], TA-DualCV [2], etc. Deep learning methods have many state-of-the-art ones [3-5], I would recommend them to be carefully discussed. And I don’t think existing imputation works all assumed random missingness, since many of them didn’t explicitly assume so, and they can impute real clinical data (naturally with mixed types of missingness) well.
- A unified framework, some work already did missingness (with/without imputation)+forcasting/classification [3-7]. If the paper claims their framework contributes from this way (e.g., lines 117-120), I would recommend the related work has to be included as baselines for a fair comparison.
- I feel IVP-VAE is a strong baseline especially in terms of AUPRC, its results should be in bold as the standard deviation (if I understand correctly since the table 1 doesn’t have a notion after ±) from multiple runs indicates the un-distinguishable performance between IVP-VAE and Mamba-IVP.
Minor issues:
- Missing proper citations for introduction part that refers to important existing work. E.g. MIMIC-IV, and the entire part of discussing prior imputation works such as GRU-D et al.
- I understand the page limitations, but related work at least should be placed in main content for self-containance
- Seems a typo: line 256: “a multi-layer perceptron (ODE)” -> “a multi-layer perceptron (MLP)”

[1] 3D-MICE: integration of cross-sectional and longitudinal imputation for multi-analyte longitudinal clinical data. Journal of the American Medical Informatics Association, 25(6), 645-653.

[2] Reconstructing missing ehrs using time-aware within-and cross-visit information for septic shock early prediction. In 2022 IEEE 10th International Conference on Healthcare Informatics (ICHI) (pp. 151-162). IEEE.

[3] Diffusion-based Time Series Imputation and Forecasting with Structured State Space Models, TMLR 2023

[4] Missing value imputation methods for electronic health records." IEEE Access 11 (2023)

[5] Kowsar, Ibna, Shourav B. Rabbani, and Manar D. Samad. "Attention-Based Imputation of Missing Values in Electronic Health Records Tabular Data." 2024 IEEE 12th International Conference on Healthcare Informatics (ICHI). IEEE, 2024.

[6] Gradient Importance Learning for Incomplete Observations. 10th International Conference on Learning Representations (ICLR). 2022.

[7] Temporal Belief Memory: Imputing Missing Data during RNN Training. In In Proceedings of the 27th International Joint Conference on Artificial Intelligence (IJCAI-2018).

**Questions:**

Please see weaknesses.

---

> ### Author Response · Authors · 2025-11-22
>
> Q1:
>
> A1: We sincerely thank the reviewer for this insightful comment. We agree that the original framing could be clearer about how imputation, forecasting, and classification are unified in our setting, and we have revised the manuscript accordingly.
>
> **Clarified scope and unified framework.**
> Our goal is not imputation in isolation, but reliable *end-to-end prediction* (forecasting and mortality classification) under extreme missingness. We revised the Introduction to explicitly state that Mamba-IVP jointly handles missingness, forecasting, and classification: rather than treating imputation and prediction as separate stages, the model learns latent dynamics that are robust to missingness while being optimized for downstream tasks.
>
> **Reorganized baseline families and added metrics.**
> To make the comparison structure explicit, we group baselines into:
>
> - **Two-stage pipelines** (imputation → prediction): MissForest+GRU, SAITS+classifier, CSDI+MLP
> - **End-to-end models with missingness handling**: GRU-D, GRU-Δt, mTAN, Raindrop
> - **Continuous-time generative models**: Latent-ODE, Latent-Flow, IVP-VAE, and our Mamba-IVP
>
> **Imputation metrics.**
> To address the request for explicit imputation evaluation, we added an imputation experiment on PhysioNet 2012 in the appendix section *Comparison Between Mamba-IVP and Existing Imputation Methods*. The RMSE results are summarized below (same as in the appendix):
>
> ### **Imputation performance (RMSE) on PhysioNet 2012 (imputation-only setting)**
>
> | Method        | 30% Missing | 50% Missing | 70% Missing |
> |--------------|-------------|-------------|-------------|
> | MissForest   | 1.34        | 1.42        | 1.48        |
> | SAITS        | 0.97        | 1.00        | 1.04        |
> | CSDI         | 0.90        | 0.94        | 0.99        |
> | IVP-VAE      | 0.79        | 0.83        | 0.88        |
> | **Mamba-IVP**| **0.76**    | **0.78**    | **0.82**    |
>
> **Two-stage vs. end-to-end comparison.**
> We also explicitly evaluate two-stage (imputer → classifier) vs. end-to-end training in the appendix section *Two-Stage vs. End-to-End*. The key results (PhysioNet 2012) are:
>
> ### **Two-stage vs. end-to-end performance on PhysioNet 2012 (50 epochs; same hardware/settings)**
>
> | Method              | Type       | MSE ↓ | AUROC ↑ | Total Time (s) ↓ |
> |--------------------|------------|-------|---------|------------------|
> | MissForest → GRU   | Two-stage  | 0.792 | 0.55    | 414              |
> | SAITS → GRU        | Two-stage  | 0.743 | 0.64    | 6186             |
> | CSDI → MLP         | Two-stage  | 0.758 | 0.62    | 1820             |
> | IVP-VAE            | End-to-end | 0.567 | 0.77    | 1630             |
> | **Mamba-IVP**       | End-to-end | **0.544** | **0.80** | **1070**     |
>
> These results show that two-stage imputation pipelines, despite strong reconstruction RMSE, do not match end-to-end models in forecasting or classification.
>
> **Computational cost of IVP-VAE and others.**
> We report execution time (forward latency, epoch time, peak memory) for all major baselines, including IVP-VAE, in the appendix section *Computational Efficiency Comparison with Baseline Models*. A summary on PhysioNet 2012 is:
>
> ### **Computational efficiency on PhysioNet 2012 (same hardware/settings)**
>
> | Method       | MSE ↓ | T_forward (s) ↓ | T_epoch (s) ↓ | Peak Mem. (MB) ↓ |
> |--------------|--------|------------------|----------------|-------------------|
> | GRU-D        | 0.586 | 0.185            | 130.6          | 342               |
> | mTAN         | 0.592 | 0.243            | 195.4          | 546               |
> | Latent-Flow  | 0.586 | 0.307            | 264.7          | 720               |
> | IVP-VAE      | 0.568 | 0.012            | 8.2            | 164               |
> | **Mamba-IVP**| **0.542** | **0.007**   | **5.2**        | **245**           |
>
> Together, these additions clarify the connection between imputation, forecasting, and classification, and ensure that the comparison across paradigms is both fair and well-documented.

---

> ### Author Response · Authors · 2025-11-22
>
> Q2: If we say imputation approaches, statistical imputation methods include more modern ones, such as MICE, 3D-MICE, TA-DualCV, etc. Deep learning methods have many state-of-the-art ones ... existing imputation works do not all assume random missingness.
>
> A2: We thank the reviewer for highlighting these important advances. We agree that our initial related work discussion was overly condensed. We have revised the *Related Work* section in the appendix to better situate our contribution.
>
> **Expanded coverage and assumptions.**
> We now explicitly include:
>
> - **Statistical methods** such as MICE, 3D-MICE, TA-DualCV, and MissForest, noting that they typically rely on MAR/structured MAR assumptions and often struggle under extreme ICU sparsity and long block-wise gaps.
> - **Deep learning methods** such as NAOMI, GRIN, CSDI, BRITS, GP-VAE, and NRTSI, clarifying that several of these do **not** assume purely random missingness and instead model structured patterns or latent-process missingness.
>
> **Positioning Mamba-IVP.**
> We emphasize that Mamba-IVP focuses on a particularly challenging regime:
> (i) very high missing rates (up to 98%),
> (ii) long block-wise outages (2–12 hours), and
> (iii) sensor noise from aging devices.
>
> While prior methods address aspects of structured missingness, they generally optimize only reconstruction fidelity and are not designed to jointly handle block-wise missingness, noise, and continuous-time forecasting/classification at the scale required for ICU deployment.
>
> **Summary table.**
> We also included a summary table in the appendix that categorizes representative methods by approach and missingness assumption. This directly addresses the reviewer’s point that not all prior work assumes random missingness.
>
>
> ### **Categorization of imputation methods by approach and missingness assumption (as added in the appendix)**
>
> | **Category**  | **Methods**       | **Missingness Assumption**     |
> |---------------|-------------------|--------------------------------|
> | Statistical   | MICE, 3D-MICE     | MAR (Missing at Random)        |
> |               | TA-DualCV         | Time-aware MAR                 |
> |               | MissForest        | MAR/MNAR adaptive              |
> | Deep Learning | BRITS             | Time-decay patterns            |
> |               | GP-VAE, NRTSI     | Structured / Latent            |
> |               | NAOMI, GRIN       | Spatiotemporal                 |
> |               | CSDI              | Probabilistic / General        |
> |               | SAITS             | Self-attention based           |
> | Ours          | **Mamba-IVP**     | Block-wise + noise             |

---

> ### Author Response · Authors · 2025-11-22
>
> Q3: A unified framework... please include such related work as baselines for fair comparison.
>
> A3: We appreciate this suggestion. Several recently proposed “unified” frameworks are designed for different data modalities or tasks (e.g., general tabular data, images, or non-clinical time series), and do not provide readily usable implementations for irregular ICU EHR data. Where prior work provides code and is applicable to multivariate irregular time series (e.g., Latent-ODE, Latent-Flow, IVP-VAE, CSDI), we have included them as baselines under the appropriate paradigm (end-to-end generative modeling or imputation). For frameworks that are conceptually related but technically incompatible with our clinical setting, we discuss them in the extended related work section but do not treat them as baselines to avoid unfair or non-representative comparisons.

---

> ### Author Response · Authors · 2025-11-22
>
> Q4: IVP-VAE AUPRC should be bold if statistically indistinguishable; clarify what $\pm$ denotes.
>
> A4: We thank the reviewer for this careful observation. In the revised manuscript, we:
>
> - Clearly state in the caption of Table~2 that “$\pm$” denotes the standard deviation across five independent runs with different random seeds.
> - Boldface results that are statistically indistinguishable from the best performance using a paired two-tailed $t$-test with $n=5$ and $p \ge 0.05$. This includes IVP-VAE where it is not significantly worse than the top performer.
> - Ensure consistent formatting and statistical notation across all result tables.
>
>
> Q5: Missing proper citations (e.g., MIMIC-IV; prior imputation works such as GRU-D).
>
> A5: We appreciate the reviewer pointing this out. We have added or corrected citations throughout the paper, including:
>
> - MIMIC-IV, PhysioNet 2012, and eICU in the Datasets subsection.
> - GRU-D, BRITS, SAITS, CSDI, Latent-ODE, GRU-ODE-Bayes, Latent-Flow, and IVP-VAE in the baseline and related work sections.
>
> This brings the manuscript into line with standard citation practices.
>
>
>
> Q6: Related work should be in the main content for self-containment.
>
> A6: We agree. In the revised version, we have:
>
> - Added a concise *Related Work* section in the main paper (Section~2) summarizing representative statistical, deep learning, and generative approaches most relevant to our setting.
> - Retained an extended survey in the appendix (Appendix~Related Work) for readers interested in a more comprehensive discussion.
>
> This strikes a balance between self-containment and page constraints.
>
>
>
>
> Q7: Typo: “a multi-layer perceptron (ODE)” $\rightarrow$ “a multi-layer perceptron (MLP)”.
>
> A7: Corrected in the Method section (EBT subsection). We now refer to the function $f_\theta$ as “a multi-layer perceptron (MLP)” when instantiated in that form.

---

> ### Author Response · Authors · 2025-11-26
>
> Dear Reviewer,
>
> We hope the above clarifications and the additional experiments in the revised draft sufficiently addressed your concerns. If you are satisfied, we kindly request you to consider updating the score to reflect the newly added results and discussion. We remain committed to addressing any remaining points you may have during the discussion phase.
>
> Best regards,
>
> The authors of Paper 4935

---

> ### Author Response · Authors · 2025-11-27
>
> Dear Reviewer h4Gc,
>
> We truly appreciate your time and effort in reviewing our manuscript. The other reviewers have already responded to the revised version and indicated that the updates have addressed their previous concerns. We sincerely hope to receive your feedback as well, as your comments are very important for the final assessment of our work. Please let us know if any additional clarification is needed.
>
> Thank you again for your valuable time and consideration.
>
> Best regards,
>
> The authors of Paper 4935

---

### Meta-Review · Area_Chair_p82H · 2026-01-16

**Summary:**

The paper proposes a Mamba-based encoder-decoder with a state-space generative model to handle missing data in EHRs. The method is evaluated on time series forecasting and mortality prediction, improving on standard metrics by around 3% over s.o.t.a.

I based my assessment on the following concerns, derived from the reviews:

W1. Ambiguity in terms of the methodological gap this paper covers. Is the focus on imputation or forecasting with missingness?
W2. Results of IVP-VAE seem to be very close to the proposed method. The baselines are limited.
W3. Novelty: reviewer YMkW indicated that "Mamba-IVP is basically an IVP-VAE with a Mamba autoencoder".

Other issues raised concerned clarifications on the technical details and requests to improve the text/figures. For instance, one reviewer believed that the datasets were limited to sepsis patients, which they are not. Another request was for the refining of Figure 1 and further description of the training procedure. These concerns were, for the most part, adequately addressed by the author response.

**Reviewer Concerns:**

W1. The authors clarified the goal is end-to-end prediction, they out baselines into 3 categories (separate stages for imputation and prediction, end-to-end models that handle missingness and continuous-time generative models). Some of these baselines are old, however, for instance GRU-D came out in 2016, Raindrop in 2022, Latent ODE in 2019. Even IVP-VAE, which came out in 2023, has been superseded (see papers citing it on Google Scholar). Also, there are papers like Latent Flow Transformer and Flow Matching for Missing Data Imputation, which might have at least been mentioned, if not compared against. So this problem of positioning with respect to related work remains.

W2. The authors have provided results on the PhysioNet dataset where they show the proposed method has more of an improvement than contenders, especially two-stage methods, in an attempt to "clarify the connection between imputation, forecasting, and classification". This experiment does add evidence in support of their claims concerning two-stage vs end-to-end, however, it doesn't address the fact that a limited number of contenders are being considered.

W3. The response did not completely alleviate the reviewer's concern, however, the reviewer did indicate that "bidirectional dynamics and theoretical analysis may be sufficient".


All things considered, the method is neither sufficiently novel, nor sufficiently apt in terms of performance to be ready for acceptance at ICLR. The theoretical analysis might be a valuable contribution (though not enough in its current form), so for the next version the authors might choose to expand upon it.

**Reviewer Scores:**

I have no way of knowing how the reviewers would have changed their scores.

---

### Decision · Program_Chairs · 2026-01-26

Reject